# Cardiac pathologies in mouse loss of imprinting models are due to misexpression of H19 long noncoding RNA

Ki-Sun Park[1†‡], Beenish Rahat[1†], Hyung Chul Lee[1], Zu-Xi Yu[2], Jacob Noeker[1], Apratim Mitra[1], Connor M Kean[1], Russell H Knutsen[3], Danielle Springer[4], Claudia M Gebert[1], Beth A Kozel[3], Karl Pfeifer[1]*

[1]Division of Intramural Research, Eunice Kennedy Shriver National Institute of Child Health and Human Development, National Institutes of Health, Bethesda, United States; [2]Pathology Core, National Heart Lung and Blood Institute, National Institutes of Health, Bethesda, United States; [3]Laboratory of Vascular and Matrix Genetics, National Heart Lung and Blood Institute, National Institutes of Health, Bethesda, United States; [4]Murine Phenotyping Core, National Heart Lung and Blood Institute, National Institutes of Health, Bethesda, United States

*For correspondence:
pfeiferk@mail.nih.gov

†These authors contributed equally to this work

Present address: ‡Division of Clinical Medicine, Korea Institute of Oriental Medicine, Daejeon, South Korea

Competing interest: The authors declare that no competing interests exist.

**Abstract** Maternal loss of imprinting (LOI) at the *H19/IGF2* locus results in biallelic *IGF2* and reduced *H19* expression and is associated with Beckwith–-Wiedemann syndrome (BWS). We use mouse models for LOI to understand the relative importance of *Igf2* and *H19* mis-expression in BWS phenotypes. Here we focus on cardiovascular phenotypes and show that neonatal cardiomegaly is exclusively dependent on increased *Igf2*. Circulating IGF2 binds cardiomyocyte receptors to hyper-activate mTOR signaling, resulting in cellular hyperplasia and hypertrophy. These *Igf2*-dependent phenotypes are transient: cardiac size returns to normal once *Igf2* expression is suppressed postnatally. However, reduced *H19* expression is sufficient to cause progressive heart pathologies including fibrosis and reduced ventricular function. In the heart, *H19* expression is primarily in endothelial cells (ECs) and regulates EC differentiation both in vivo and in vitro. Finally, we establish novel mouse models to show that cardiac phenotypes depend on *H19* lncRNA interactions with *Mirlet7* microRNAs.

## Introduction

There are 100–200 imprinted genes in mammals. These genes are organized into discrete clusters where monoallelic expression is dependent on a shared regulatory element known as the *Imprinting Control Region* (*ICR*) (*Barlow and Bartolomei, 2014*). Imprinted genes are frequently involved in human disease and developmental disorders (*Eggermann et al., 2015*; *Feinberg and Tycko, 2004*; *Horsthemke, 2014*; *Kalish et al., 2014*; *Peters, 2014*). Sometimes, these diseases are due to inactivating point mutations of the only transcriptionally active allele. Alternatively, imprinting diseases are caused by disruption of ICR function, leading to mis-expression of all genes in the cluster.

One imprinted cluster is the *IGF2/H19* locus on human chromosome 11p15.5. Imprinting in this >100 kb region is determined by the *H19ICR*, located just upstream of the *H19* promoter (*Kaffer et al., 2000*; *Thorvaldsen et al., 1998*). As described in *Figure 1A*, the *H19ICR* organizes the locus such that transcription of the *IGF2* (*Insulin-like Growth Factor 2*) and *H19* genes are restricted to the paternal and maternal chromosomes, respectively (*Ideraabdullah et al., 2008*; *Murrell, 2011*; *Yoon*

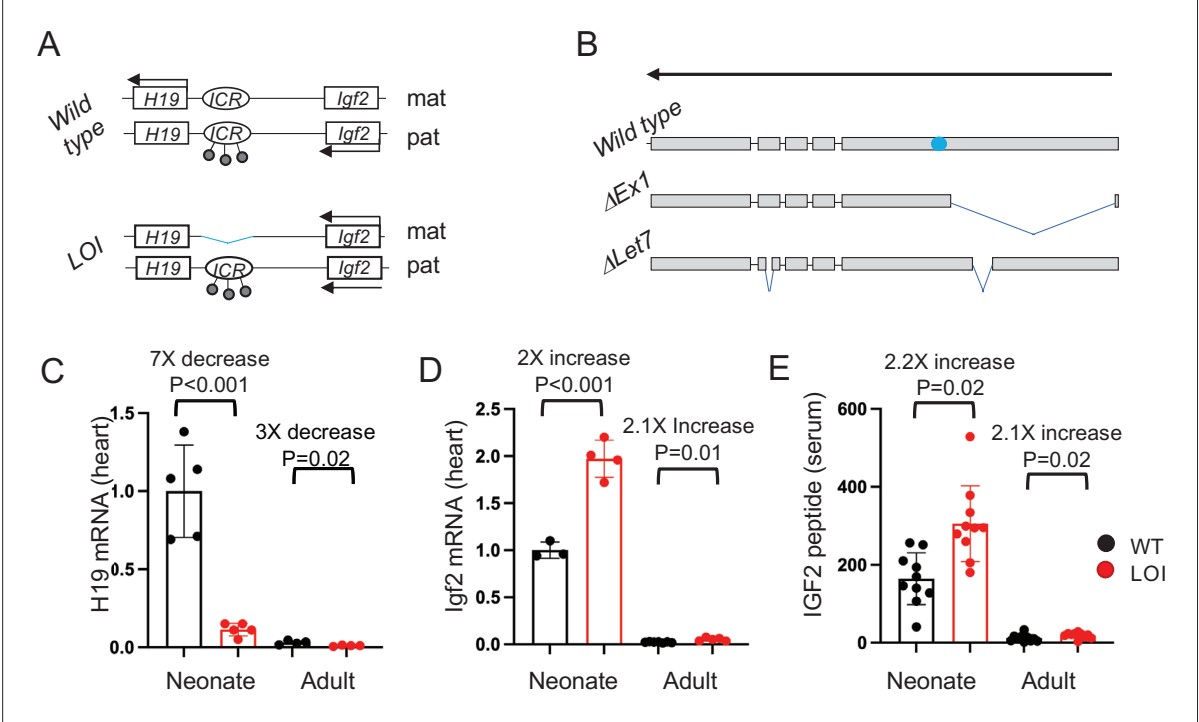

**Figure 1.** *The H19/Igf2* locus. (**A**) Schematic of maternal (mat) and paternal (pat) chromosomes in wild-type and in loss of imprinting (LOI) mice. Gene expression is indicated by horizontal arrows. In wild-type mice, the paternal copy of the imprinting control region (ICR) is inactivated by DNA methylation (filled lollipops). In LOI patients, inappropriate inactivation of the maternal ICR typically occurs due to microdeletion or to inappropriate DNA methylation. In the mouse LOI model, the maternal ICR is inactivated by deletion. (**B**) Schematic of *wild-type*, *ΔEx1*, and *ΔLet7 H19* alleles. *H19* exons 1–5 are shown as filled rectangles. *ΔEx1* is a 700 bp deletion at the 5′ end of exon 1. *ΔLet7* was constructed for this study by simultaneous deletion of *Mirlet7* binding sites in *H19* exons 1 and 4. The blue oval identifies coding sequences for *Mir675-3p* and *-5* p. Arrowheads show the direction of transcription. (**C–E**). Maternal loss of imprinting results in reduced *H19* lncRNA and 2× doses of *Igf2*. (**C, D**) Hearts were isolated from wild type (WT) or from *H19^ΔICR^/H19^+^* (LOI) littermates at postnatal day 2 or at 2 months. RNAs were extracted, analyzed by qRT-PCR, normalized to GAPDH, and then normalized to levels observed in wild-type neonates. Despite the dramatic postnatal repression, *H19* expression in adults remains substantial as *H19* lncRNA is in the top 10-percentile of all RNAs. (**C**) IGF2 peptide levels in serum were measured by ELISA. Statistical significance was evaluated with Student's t-test type 2.

The online version of this article includes the following source data for figure 1:

**Source data 1.** Maternal Loss of Imprinting (LOI) results in decreased Igf2 and increased H19 expression.

*et al., 2007*). (Note that in medical genetics, the *H19ICR* is also known as Imprinting Center one or IC1).

*IGF2* encodes a peptide hormone that binds to and activates the insulin receptor (InsR) and insulin-like growth factor one receptor (Igf1R) kinases to promote cell growth and proliferation in many cell types including cardiomyocytes (*Bergman et al., 2013*; *Geng et al., 2017*; *Li et al., 2011*; *Wang et al., 2019*). In contrast, the functional product of the *H19* gene is a 2.3 kb long non-coding RNA whose biochemical functions remain controversial (*Brannan et al., 1990*; *Gabory et al., 2010*). Reported roles for the *H19* lncRNA include (1) acting as the precursor for microRNAs (*Mir675-3p* and *Mir675-5p*) (*Cai and Cullen, 2007*; *Keniry et al., 2012*), (2) regulating the bioavailability of *Mirlet7* (let-7) microRNAs (*Gao et al., 2014*; *Geng et al., 2018*; *Kallen et al., 2013*; *Li et al., 2015*), (3) interacting with p53 protein to reduce its function (*Hadji et al., 2016*; *Park et al., 2017*; *Peng et al., 2017*; *Yang et al., 2012*; *Zhang et al., 2019*; *Zhang et al., 2017*), and (4) regulating DNA methylation to thereby modulate gene expression (*Zhou et al., 2019*; *Zhou et al., 2015*).

In humans, disruption of the maternally inherited *H19ICR* results in biallelic *IGF2* along with reduced *H19* expression and is associated with the developmental disorder, Beckwith–Wiedemann syndrome (BWS) (*Jacob et al., 2013*). BWS is a fetal overgrowth disorder but the specific manifestations of overgrowth vary between patients. Cardiomegaly is a common newborn presentation but typically resolves without treatment. Cardiomyopathies are rarer and include ventricular dilation, valve/septal

defects, fibrotic and rhabdomyoma tumors, and vascular abnormalities (*Cohen, 2005*; *Descartes et al., 2008*; *Drut et al., 2006*; *Elliott et al., 1994*; *Greenwood et al., 1977*; *Knopp et al., 2015*; *Longardt et al., 2014*; *Ryan et al., 1989*; *Satgé et al., 2005*). BWS incidence correlates with artificial reproductive technologies (ART) (*DeBaun et al., 2003*; *Gicquel et al., 2003*; *Halliday et al., 2004*; *Hattori et al., 2019*; *Johnson et al., 2018*; *Maher et al., 2003*; *Mussa et al., 2017*), and among BWS patients, the frequency of heart defects is higher in those born via ART (*Tenorio et al., 2016*).

We have generated a mouse model that recapitulates the molecular loss of imprinting (LOI) phenotypes of BWS (*Figure 1A*; *Srivastava et al., 2000*). That is, deletion of the *H19*ICR on the maternal chromosome results in biallelic *Igf2* and reduced levels of *H19*. In this study, we show that the LOI mouse model displays cardiovascular defects seen in BWS patients. Genetic and developmental analyses indicate that mis-expression of *Igf2* and *H19* act independently on distinct cell types to cause the cardiac phenotypes. During fetal development, increased circulating IGF2 activates AKT/mTOR pathways in cardiomyocytes resulting in cellular hypertrophy and hyperplasia. This neonatal hypertrophy is transient, non-pathologic, and unaffected by the presence or absence of a functional *H19* gene. However, loss of *H19* lncRNA results in cardiac fibrosis and hypertrophy and a progressive cardiac pathology in adult animals. In both neonatal and adult hearts, *H19* lncRNA expression is restricted to endothelial cells (ECs). In vivo, loss of *H19* results in high incidence of ECs that co-express endothelial and mesenchymal markers. Similarly, primary cardiac endothelial cells can be driven toward a mesenchymal phenotype by manipulating *H19* expression levels. Thus, this research identifies a novel developmental role for the *H19* lncRNA in regulating cardiac endothelial cells. In fact, this role for *H19* in restricting endothelial cell transitions in the heart is unexpected given previous analyses of *H19* function in vitro in transformed cell lines. Finally, we describe structure–function analyses in two novel mouse models (*Figure 1B*) and show that *H19* lncRNA acts by regulating *Mirlet7* bioavailability.

## Results
### Defective structure and function in hearts from mice with H19/Igf2 maternal LOI

Wild-type and LOI mice were generated by crossing $H19^{\Delta ICR}/H19^+$ with wild-type C57Bl/6 J males. (See *Figure 1A* for a description of the *H19ΔICR* allele). In mice (as in humans), maternal LOI results in biallelic (2 ×) expression of *Igf2* and reduced levels of *H19* RNA (*Figure 1C–E*). Hearts isolated from P1 LOI mice display cellular hyperplasia and cellular hypertrophy. Hyperplasia is indicated by increased staining for Ki-67 (a marker for cell proliferation) in tissue sections (*Figure 2A,C*) and by increased levels of Ki-67 and of cyclins E1 and D1 in protein extracts (*Figure 2D*, *Figure 2—figure supplement 1A*). Cellular hypertrophy is demonstrated by measuring surface areas of primary cardiomyocytes isolated from wild-type and LOI neonates (*Figure 2E,F*). Apart from their increased size, neonatal LOI hearts do not display any obvious pathologies. For example, we did not see increased fibrosis or expression of protein markers associated with heart disease. Furthermore, by 2 months of age, we were unable to distinguish LOI mice by cardiomegaly as measured by heart weight/tibia length ratios: WT = 6.7 ± 0.3 mg/mm (N = 8), LOI = 6.8 ± 0.3 mg/mm (N = 13), p=0.79 (Student's t-test).

We continued to monitor cardiovascular phenotypes in LOI and wild-type mice until 19 months of age. By 6 months, LOI mice displayed cardiac hypertrophy as measured by a 28 % increase in heart weight/tibia length ratios (wild type = 10.0 ± 1.7 mg/mm, N = 8; LOI = 12.8 ± 0.2 mg/mm, N = 10; p=0.005). Transverse sections revealed increased fiber diameter in LOI hearts (wild type = 10.2 ± 0.7 µm; LOI = 14.4 ± 0.8 µm; p=0.007) (*Figure 3A*). Cardiac hypertrophy is often a poor prognostic sign and is associated with most forms of heart failure (*Heinzel et al., 2015*; *Vakili et al., 2001*). However, hypertrophy can also be physiologic (*McMullen and Jennings, 2007*; *Shimizu and Minamino, 2016*). The hypertrophy in LOI mice might be considered pathologic based on increased levels of ANP, Myh7, cleaved Caspase-3, cleaved Caspase-7, and cleaved PARP proteins as well as decreased levels of Serca2 protein in all LOI mice by 1 year of age (*Figure 3B*, *Figure 3—figure supplement 1*; *Mitra et al., 2013*; *van Empel et al., 2005*). Finally, both interstitial and perivascular fibrosis are prominent in LOI animals by 6 months of age (*Figure 3C,D*).

*Table 1* summarizes echocardiography phenotypes from 13-month-old mice. Left ventricles (LV) from LOI mice are dilated (as measured by increased LV volumes at both systole and diastole), mildly hypertrophic (as measured by increased wall thickness, LVAW diastole and LVPW diastole), and

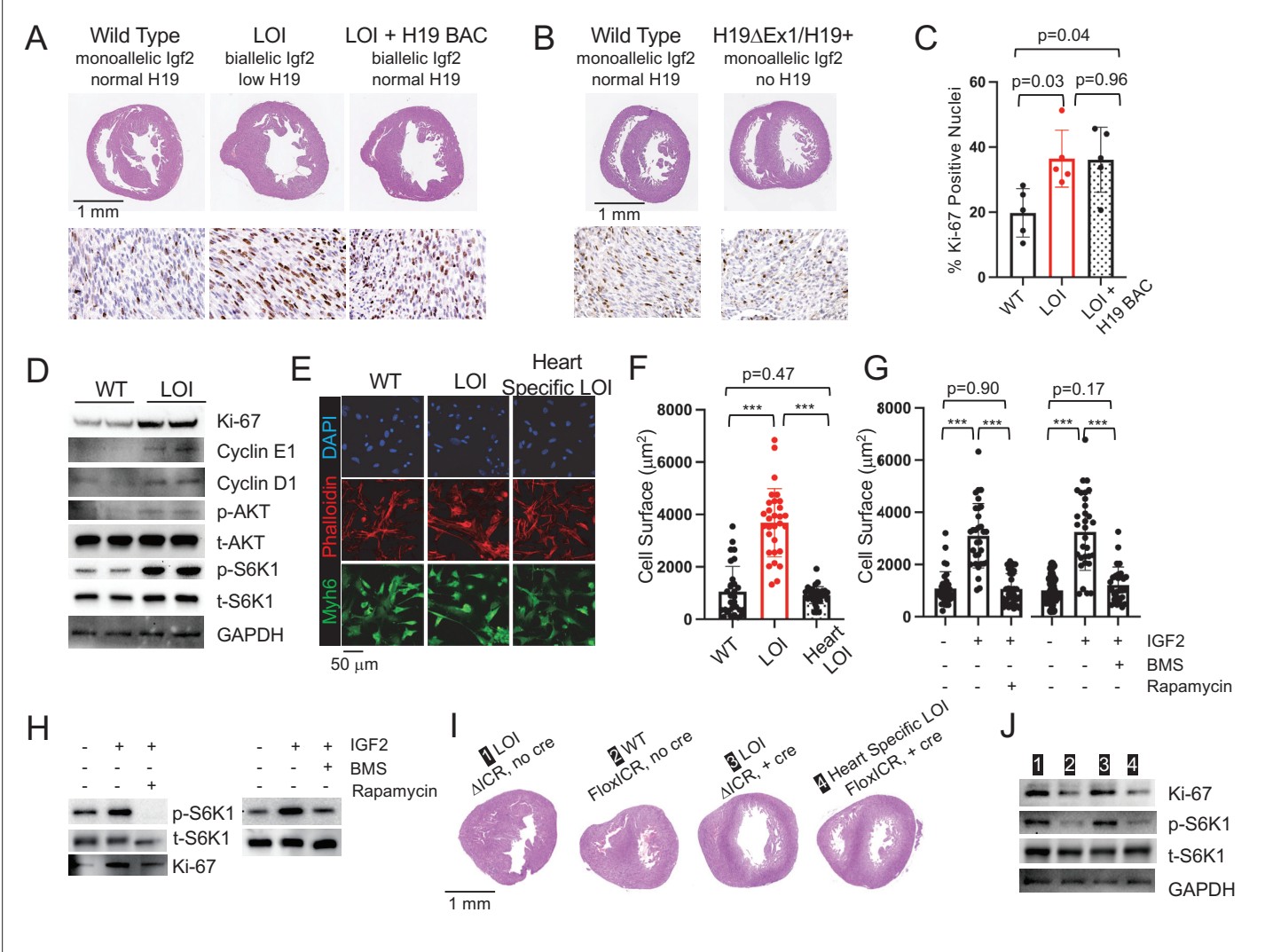

**Figure 2.** Cardiac hypertrophy in neonatal LOI mice is mediated by circulating IGF2 activation of AKT/mTOR signaling in cardiomyocytes and is independent of *H19* gene function. (**A, B**) Heart morphology in wild-type, LOI, and LOI+ H19 BAC littermates (**A**) or in wild-type and *H19*-deficient littermates (**B**) Top panels, transverse sections were taken from fixed hearts at 200 mm from the apex. Bottom panels, Ki-67 (brown stain) is a marker for cell proliferation. LOI, *H19^{ΔICR}/H19^+*; LOI+ BAC, *H19^{ΔICR}/H19^+* that also carry a 140 kb Bacterial Artificial Chromosome transgene that restores normal *H19* expression (***Figure 2—figure supplement 2***). Notice the thickened walls, misshaped right ventricles, and high levels of Ki-67 expression in LOI and in LOI+ BAC transgenic neonates. (**C**) Quantitation of Ki-67 expression as assayed in panel A. (N = 5). (**D**) Immunoblot analyses of heart extracts prepared from wild-type and LOI littermates. LOI hearts show increased levels of proliferation markers, Ki-67, Cyclin EI, and Cyclin D1 and also increased levels of phosphorylated AKT and S6K1 (a target of mTORC1). See ***Figure 2—figure supplement 1A*** for quantifed results. (**E, F**) Cardiomyocyte cellular hypertrophy in LOI animals is cell non-autonomous. Primary cardiomyocyte cultures were prepared from wild type, LOI, and from littermates carrying an *ICR* deletion only in cardiomyocytes (see below). Cells were cultured overnight, stained for MYH6 (to identify cardiomyocytes) and Phalloidin (to facilitate measurement of surface areas). For each culture (N = 5 per genotype), at least 30 cells were measured. (**G, H**) Exogenous IGF2 peptide induces cellular hypertrophy in wild-type cardiomyocytes through mTOR pathways. Primary cardiomyocytes were prepared from wild-type neonates and cultured overnight with IGF2 before measurement of cell surface area (**G**) or preparation of protein extracts for immunoblotting. (**H**) The effect of increased IGF2 is prevented by treatment with BMS 754807 or with Rapamycin. BMS inhibits IgfR1 and Ins2 receptor kinases (***Carboni et al., 2009***). Rapamcyin blocks a subset of mTOR activities (***Li et al., 2014***). See ***Figure 2—figure supplement 1B*** for quantified results. (**I, J**) LOI phenotypes in cardiomyocytes are cell non-autonomous. *H19^{ICRflox}/H19^{ΔICR}* females were crossed with males carrying the *Myh6-Cre* transgene to generate four kinds of pups: *H19^{ΔICR}/H19^+* (#1) and *H19^{ΔICR}/H19^+ Myh6* Cre (#3) will display LOI in all cell types; *H19^{ICRflox}/H19^+* (#2) will display wild-type expression patterns for *Igf2* and *H19*; and *H19^{ICRflox}/H19^+ Myh6* Cre mice will show LOI only in cardiomyocytes. Hearts were analyzed for cellular hypertrophy (**E**), megacardia and hyperplasia (**I**), and protein expression (**J**). See ***Figure 2—figure supplement 1C*** for quantified results of protein expression. In all assays, *H19^{ICRflox}/H19^+ Myh6* Cre mice were highly similar to their wild-type littermates and distinct from the congenital LOI littermates. *p<0.05; ***p<0.001 (Student's t-test). LOI, loss of imprinting (*H19^{ΔICR}/H19^+*).

*Figure 2 continued on next page*

*Figure 2 continued*

The online version of this article includes the following source data and figure supplement(s) for figure 2:

**Source data 1.** Analyses of LOI phenotype in neonatal mice.

**Figure supplement 1.** *LOI-dependent changes in protein expression in neonatal mice – quantitated western blots.*

**Figure supplement 2.** The H19 BAC transgene restores H19 expression in LOI hearts.

**Figure supplement 3.** The H19ΔEx1 deletion reduces H19 lncRNA but has only only a small impact on Mir675-3p and -5 p accumulation.

**Figure supplement 4.** *Tissue-specific expression of H19.*

show diminished function (as measured by reduced ejection fractions, % EF). LOI mice showed large increases in velocity and turbulence of blood flow from the LV outflow tract. Finally, major vessel lumen diameters (measured at the aortic arch and the first brachial arch) were >30% larger in LOI mice.

Scatterplots of echocardiography data from 13 month animals show that most LOI phenotypes are heterogeneous and are not normally distributed (*Figure 3—figure supplement 2*). Rather phenotypes for volume, mass, ejection fraction, and outflow tract velocity and turbulence are all bimodal: 6–7 animals display mild phenotypes, and 3–4 animals display extreme pathologies (*Figure 3—figure supplement 2A–D*). The only exception to this pattern is seen in arterial diameter phenotypes. In this case, the variance among LOI animals is low (like their WT cohorts) and all the LOI animals display a pathologic phenotype (*Figure 3—figure supplement 2E,F*).

*Supplementary file 1* summarizes correlations between the various phenotypes identified by echocardiography. Cardiac function as measured by ejection fraction is inversely correlated with LV volume (RR = 0.81). However, function correlates only moderately with wall thickness (RR = 0.53) and not at all with outflow tract defects (RR = 0.02), or with aortic diameter (RR < 0.01). Thus, LOI associated phenotypes are not uniformly penetrant. Rather, each mouse presents a distinct array of defects. The only invariant is that all LOI mice have arterial diameters larger than their wild-type counterparts.

The right-hand columns in *Table 1* summarize echocardiography results from the same mice at 16 months of age. We observed the same ventricular abnormalities: reduced ejection fraction, increased chamber size, and increased wall thickness. However, on scatter plots, we see that wild-type and mutant animals now show non-overlapping phenotypes, consistent with the idea that ventricular failure is progressing in LOI mice (*Figure 3—figure supplement 2*). Note that the LOI mouse with the poorest function at 13 months (25 % EF) died prior to this second analysis.

Finally, in vivo analyses at 19 months identified significant pathological reductions in both systolic blood pressure (WT = 105 ± 2, LOI = 93 ± 3, p=0.01) and pulse pressure (WT = 37 ± 1, LOI = 28 ± 1, p<0.001) in mutant mice (*Figure 3—figure supplement 3*). These data confirm that *H19/Igf2* LOI has a substantial effect on cardiovascular function.

As described above, increased artery diameter is a phenotype where by 13 months, LOI and WT mice sorted into phenotypically distinct cohorts. This suggested that abnormal blood vessel structure might be a relatively primary defect. We focused additional attention to this phenotype and measured outer diameters of isolated ascending aorta and carotid arteries in response to applied pressures on a pressure myograph (*Figure 3E*, *Figure 3—figure supplement 4A*). Arteries from LOI mice are larger in diameter across all applied pressures. Moreover, across normal physiological pressure ranges (75–125 mmHg) arteries from mutant mice are more sensitive to changes in pressure and lumens reach their maximum diameter at lower pressures. They are appropriately distensible at low (elastic) pressures but are stiffer than WT vessels over higher pressure intervals, including most physiologic pressures (*Figure 3F*, *Figure 3—figure supplement 4B*).

In sum, *H19/Igf2* LOI in mice results in transient neonatal cardiomegaly and then a progressive cardiomyopathy. Note that results shown in *Figure 3* and in *Table 1* describe comparisons of age-matched male mice. Adult LOI females consistently showed relatively weak phenotypes and p-values were not significant (data now shown). However, neonatal hypertrophy and hyperplasia occurs in both male and female pups. This apparent paradox was the first clue that the relationship between the neonatal hypertrophy and the adult disease phenotypes was not straightforward.

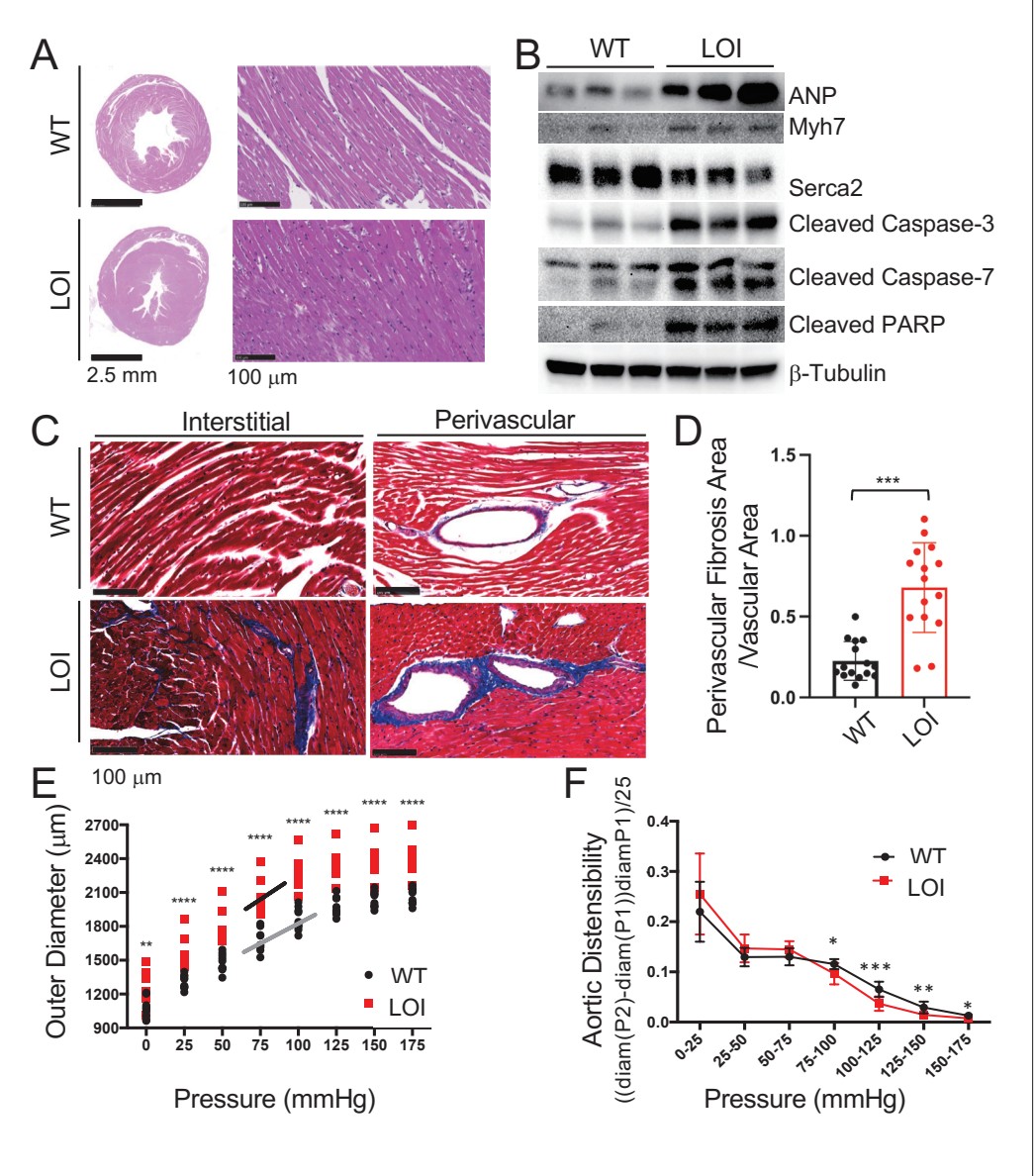

**Figure 3.** Cardiomyopathies in adult LOI mice. (**A**) Transverse sections were collected midway along the longitudinal axis from hearts collected from 6-month-old wild-type (WT) and LOI mice and stained with hematoxylin and eosin. (**B**) Immunoblot analyses of whole heart extracts prepared from 1 year WT and LOI mice. Note the altered expression of ANP (Atrial Natriuretic Peptide), Myh7 (Myosin Heavy Chain 7), Serca2 (Sarco/endoplasmic reticulum Ca++ ATPase), Cleaved Caspase-3, and Cleaved Poly ADP Ribose Polymerase (PARP). β-Tubulin is a loading control. See *Figure 3—figure supplement 1* for quantified data. (**C, D**) Masson's trichrome staining of sections described in (**A**). Red, muscle fibers; blue, collagen. Sections from five wild-type and five LOI animals were used to calculate fibrosis. Statistical significance was evaluated with Student's t-test type 2. (**E, F**) Ascending aortas were isolated from 10 wild-type and 8 LOI mice at age 20 months and pressure-diameter curves generated. (**E**) Increased diameters across a wide range of applied pressures. (**F**) Increased segmental distensibility across physiologically relevant pressures. Data were analyzed by two-way repeated measure ANOVA. Analyses of carotid arteris are described in *Figure 3—figure supplement 4*.

The online version of this article includes the following source data and figure supplement(s) for figure 3:

**Source data 1.** Analyses of cardiomyopathies in adult LOI mice.

**Figure supplement 1.** LOI-dependent changes in protein expression in adult mice – quantitated western blots.

**Figure supplement 2.** Echocardiography measures from 11 wild-type and 10 LOI mice at 13 months and for 11 wild-type and 9 LOI mice at 16 months.

*Figure 3 continued on next page*

*Figure 3 continued*

**Figure supplement 3.** Decreased systolic and pulse pressures in LOI mice.

**Figure supplement 4.** Increased vessel diameter and distensibility in carotid arteries isolated from 10 wild-type and 8 LOI mice at 16 months.

## Hypertrophy and hyperplasia in neonatal LOI mice is dependent on hyperactivation of mTOR/AKT signaling by increased dosage of IGF2 peptide

To understand the specific roles for mis-expression of *Igf2* and of *H19* in neonatal cardiomegaly, we performed two genetic analyses. First, we rescued *H19* expression in an LOI background by introducing a 140 kb H19 Bacterial Artificial Chromosome (H19 BAC) (*Kaffer et al., 2001*; *Kaffer et al., 2000*; *Figure 2—figure supplement 2*) but still saw cardiomyocyte hypertrophy and hyperplasia in neonates (*Figure 2A,C*). Second, we tested the effect of removing *H19* in a background where *Igf2* remains monoallelic by comparing $H19^{\Delta Ex1}/H19^+$ (*Figure 1B*) with wild-type littermates. (Reduced expression of *H19* lncRNA in $H19^{\Delta Ex1}/H19^+$ is shown in *Figure 2—figure supplement 3A*.) Loss of *H19* lncRNA does not result in neonatal cardiomyocyte hypertrophy or hyperplasia (*Figure 2B*). Altogether, we conclude that loss of *H19* lncRNA does not contribute to neonatal hypergrowth. Rather, this neonatal hypertrophy is dependent only upon biallelic (2 × dosage) *Igf2* transcription.

IGF2 peptide works by binding and activating InsR and IgfR kinases and mTOR/AKT signaling is a known downstream target of these receptor kinases (*Bergman et al., 2013*). In addition, studies document the role of AKT/mTOR signaling in cardiomyocyte cell division and hypertrophy (*Geng et al., 2017*; *Li et al., 2011*; *Sciarretta et al., 2014*; *Shen et al., 2020*; *Wang et al., 2019*). Consistent with a critical role for AKT/mTOR signaling in LOI-dependent neonatal hypertrophy, hearts from LOI neonates show increased levels of phosphorylated AKT and of phosphorylated S6K1, a downstream marker for mTORC1 activity (*Figure 2D*, *Figure 2—figure supplement 1A*). Moreover, the LOI cellular hypertrophy and pAKT hyperactivation phenotypes can be phenocopied by treatment of wild-type primary cardiomyocytes with IGF2 peptide. However, IGF2 action is blocked by BMS-754807, a specific inhibitor of the receptor kinase, or by treatment with rapamycin, an mTOR signaling pathway inhibitor (*Figure 2G and H*; *Figure 2—figure supplement 1B*).

*Igf2* is widely expressed in the embryo. In fact, expression of *Igf2* is low in the heart relative to other tissues, especially liver and skeletal muscle (*Figure 2—figure supplement 4*), which are believed to be the major source of circulating IGF2 peptide. Within the heart, IGF2 originates in both cardiomyocytes but also in endothelial cells (*Shen et al., 2020*). To assess the role of biallelic expression of *Igf2* in the cardiomyocytes themselves, we crossed $H19^{\Delta ICR}/H19^{ICRflox}$ females with $H19^+/H19^+$ males carrying the Myh6-Cre transgene. $H19^{ICRflox}$ is an allele where the H19ICR is flanked with *loxP* sites, so that cre recombination results in deletion of the *ICR* (*Srivastava et al., 2000*). We used PCR analyses to demonstrate that the transgene drives efficient ICR deletion in the heart but not in other tissues tested (skeletal muscle, liver, kidney, brain, thymus, spleen, and lung). Our cross generated wild-type mice ($H19^{ICRflox}/H19^+$) and two kinds of LOI controls ($H19^{\Delta ICR}/H19^+$;+ Myh6 Cre and $H19^{\Delta ICR}/H19^+$) that we compared with experimental mice that had cardiomyocyte-specific LOI ($H19^{ICRflox}/H19^+$;+ Myh6 Cre). Cardiomyocyte-specific ICR deletion does not cause hypertrophy. Rather, $H19^{ICRFlox}/H19^+$ Myh6 Cre mice were indistinguishable from their wild-type littermates (*Figure 2E,F,I,J*; *Figure 2—figure supplement 2C*).

Note that our data demonstrate that the effects of LOI on neonatal cardiomyocytes are cell non-autonomous but do not rule out a cell autonomous role in other cell types, including cardiac endothelium.

Cardiac disease in adults is dependent only on loss of H19 lncRNA expression. Biallelic Igf2 and the resultant hypertrophy in neonatal hearts are not relevant to the adult LOI phenotype.

While the *H19* BAC transgene does not prevent neonatal cardiomegaly, it does successfully prevent adult pathologies. That is, hearts from 6 month LOI mice carrying the *H19* transgene are not enlarged as determined by heart weight/tibia length ratios (LOI = 12.9 ± 0.6 mg/mm, N = 3; LOI+ H19 BAC transgene = 11.5 ± 0.7, N = 3; p<0.05), are not fibrotic (*Figure 4A,B*), and do not express cardiomyopathy markers (*Figure 4C*; *Figure 4—figure supplement 1A*). Thus, loss of *H19* is necessary to induce LOI cardiomyopathies.

**Table 1.** Echocardiograph of wild-type (WT) and loss of imprinting (LOI) mice at 13 and at 16 months.

LV, left ventricle; EF, ejection fraction; AW, anterior wall; PW, posterior wall; OT, outflow track. p values were calculated using Student's t-test (Type 1).

| Phenotype | 13 months | | | | 16 months | | | |
|---|---|---|---|---|---|---|---|---|
| | Mean ± SEM | | | | Mean ± SEM | | | |
| | Wt (N = 11) | Loi (N = 10) | p-value | % Change | Wt (N = 11) | Loi (N = 9) | p-value | % Change |
| Heart rate (bpm) | 504 ± 12 | 499 ± 11 | 0.77 | -1 | 529 ± 20 | 540 ± 16 | 0.77 | 2 |
| LV volume systole (µl) | 24.1 ± 1.5 | 39.2 ± 4.7 | 0.01 | 63 | 28.3 ± 1.4 | 46.6 ± 1.6 | <0.001 | 65 |
| LV volume diastole (µl) | 67.9 ± 3.0 | 79.8 ± 4.6 | 0.05 | 17 | 74.2 ± 2.8 | 91.3 ± 3.3 | <0.001 | 23 |
| LV EF (%) | 64.7 ± 1.2 | 51.9 ± 3.8 | <0.01 | −19 | 62.5 ± 0.9 | 48.6 ± 1.7 | <0.001 | −22 |
| LVAW systole (mm) | 1.40 ± 0.01 | 1.44 ± 0.02 | 0.09 | 3 | 1.39 ± 0.01 | 1.52 ± 0.04 | 0.01 | 8 |
| LVAW diastole (mm) | 0.88 ± 0.01 | 1.01 ± 0.03 | <0.01 | 15 | 0.91 ± 0.01 | 1.08 ± 0.04 | <0.001 | 20 |
| LVPW systole (mm) | 1.35 ± 0.01 | 1.39 ± 0.02 | 0.17 | 3 | 1.34 ± 0.01 | 1.42 ± 0.03 | 0.4 | 6 |
| LVPW diastole (mm) | 0.87 ± 0.02 | 0.98 ± 0.04 | 0.02 | 12 | 0.88 ± 0.02 | 1.08 ± 0.04 | <0.001 | 19 |
| LVOT mean gradient | 2.4 ± 0.2 | 6.8 ± 1.8 | 0.04 | 185 | 2.3 ± 0.2 | 4.9 ± 1.1 | 0.04 | 115 |
| LVOT mean velocity | 768 ± 30 | 1211 ± 162 | 0.02 | 58 | 750 ± 37 | 1066 ± 112 | 0.02 | 42 |
| LVOT peak gradient | 5.8 ± 0.3 | 15.7 ± 3.7 | 0.03 | 170 | 5.5 ± 0.5 | 13.3 ± 3.3 | 0.04 | 143 |
| LVOT peak velocity | 1201 ± 35 | 1870 ± 215 | 0.01 | 56 | 1158 ± 52 | 1732 ± 200 | 0.02 | 50 |
| Aorta systole (mm) | 1.65 ± 0.04 | 2.14 ± 0.10 | <0.01 | 30 | 1.70 ± 0.04 | 2.25 ± 0.12 | 0.002 | 32 |
| Aorta diastole (mm) | 1.44 ± 0.04 | 1.86 ± 0.07 | 0.001 | 30 | 1.48 ± 0.04 | 2.07 ± 0.13 | 0.002 | 40 |
| First brachial arch (mm) | 0.78 ± 0.03 | 1.06 ± 0.07 | 0.003 | 36 | 0.78 ± 0.02 | 1.09 ± 0.08 | 0.004 | 40 |

The online version of this article includes the following source data for table 1:

**Source data 1.** Analyses of heart function in wild type and LOI adult mice.

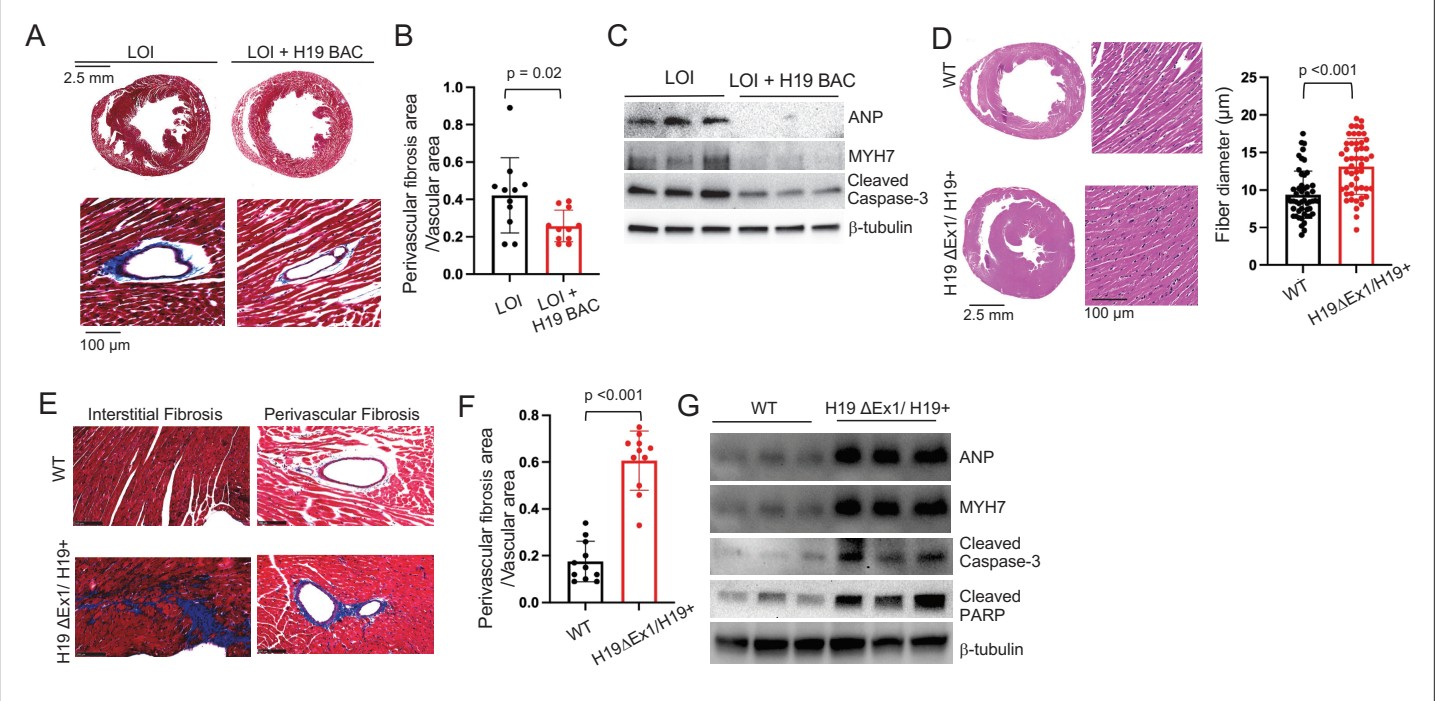

**Figure 4.** LOI pathologies in adult mice are *H19*-dependent. (**A–C**) An *H19* transgene rescues pathologies in LOI mice. Phenotypes of LOI mice or their LOI littermates that also carry an H19 Bacterial Artificial Chromosome transgene that restores wild-type levels of *H19* RNA (LOI+ H19 BAC). (**D–G**) *H19* deletion is sufficient to cause cardiac pathologies. Phenotypes in wild-type (WT) mice and in littermates carrying the *H19ΔEx1* deletion. For histology (**A, B, E, F**) hearts were isolated from 6 month old animals and transverse sections collected midway along the longitudinal axes before staining with hematoxylin and eosin (**D**) or with Masson's trichrome (**A, B, E, F**) Statistical significance was analyzed using Student's t-test type 2. For immunoblotting (**C, G**), hearts were isolated from 1 year animals and investigated for ANP, Myh7, Serca2, Cleaved Caspase-3, and Cleaved PARP. β-Tubulin is a loading control. See *Figure 4—figure supplement 1* for quantified data.

The online version of this article includes the following source data and figure supplement(s) for figure 4:

**Source data 1.** Pathologies in LOI mice are caused by decreased function of the H19 lncRNA.

**Figure supplement 1.** H19 dependent changes in protein expression in adult mice – quantitated Western blots.

We next investigated whether loss of *H19* is sufficient to induce pathologies and also investigated exactly which *H19* RNA was important. The *H19* gene encodes a 2.3 kb lncRNA which is exported to the cytoplasm but also is the precursor for microRNAs, *Mir675-5p* and *Mir675-3p* (*Cai and Cullen, 2007*). Since LOI mice show reduced levels of both the lncRNA and of *Mir675-3p* and -5 p and because the H19 BAC transgene restores expression of both the lncRNA and the *Mir675* microRNAs, these models were not helpful in determining which RNA species prevents cardiac pathology. The *H19ΔEx1* allele is a 700 bp deletion of the 5' end of exon one that leaves bases encoding the *Mir675-3p* and *Mir675-5p* intact (*Figure 1B*). This *ΔEx1* deletion does not prevent *H19* transcription but rather, reduces *H19* lncRNA levels by destabilizing the truncated transcript (*Srivastava et al., 2003*), raising the possibility that the *ΔEx1* mutation might affect only the lncRNA. In fact, we show here that levels of *Mir675-5p* and -3 p are unaltered in *H19^{ΔEx1}/H19^+* (*Figure 2—figure supplement 3*). Yet, 6-month-old *H19^{ΔEx1}/H19^+* mice display LOI cardiac pathologies including hypertrophy (*Figure 4D*), fibrosis (*Figure 4E,F*), and expression of disease markers (*Figure 4G*, *Figure 4—figure supplement 1B*). Thus, we conclude that loss of *H19* lncRNA is sufficient to induce cardiomyopathy in adult mice.

In addition to establishing the critical importance of *H19* lncRNA, these genetic experiments also uncouple neonatal hypertrophy and adult pathology: neonatal LOI+ H19 BAC mice show hypertrophy but do not develop adult pathologies, while neonatal *H19^{ΔEx1}/H19^+* have normal-sized hearts but do develop pathologies. Thus, neonatal cardiomegaly is not a risk factor for adult pathologies.

# H19 lncRNA regulates the frequency of endothelial to mesenchymal transition in mice and in isolated primary endothelial cell cultures

*H19* expression is not uniform throughout the heart but rather restricted to endothelial cells (ECs) (*Figure 5A,B*). In fetal and neonatal hearts *H19* is expressed in all endothelial cells including microvasculature. In adults, *H19* expression is patchy and patterns vary between mice but are always restricted to endocardium and endothelial cells lining major coronary vessels (*Figure 5A*). Localization was confirmed in vasculature by co-staining for both endothelial and smooth muscle markers. For example, in coronary vessels, *H19* RNA expression exclusively overlapped with endothelial specific marker von-Willebrand's factor (*Figure 5B*).

To identify a possible function for *H19* RNA, we performed transcriptomic analyses comparing RNAs isolated from wild-type and *H19*-deficient P1 hearts. Using whole heart extracts, we did not identify significant differences in gene expression. We next compared RNAs isolated from purified ECs. Hearts were dissociated into single cells using enzyme digestion and mechanical agitation and then endothelial cells were isolated based on expression of CD31 antigen. About 30,000 cells per neonatal heart were isolated to >95% purity. RNA sequencing identified 228 differentially expressed genes (DEGs) with adjusted p-values of <0.1, including 111 upregulated and 117 downregulated transcripts (*Figure 5C*). GO analysis for biological, cellular, and molecular pathways gives evidence for a change in cellular identity (*Figure 5—figure supplement 1*). Specifically, enriched biological pathways included positive regulation of mesenchymal cell proliferation, positive regulation of endothelial cell migration, and cell adhesion (n = 36, p-adj = 0.003). Cellular pathways showed enrichment for genes coding for extracellular matrix (n = 42, p-adj = 1.83E-10). Enriched molecular function categories include extracellular matrix binding and TGFβ binding (n = 7, p-adj = 0.0001; n = 5, p-adj = 0.1), as well as other pathways that are especially active during endothelial to mesenchymal transition (EndMT). EndMT is not an identifiable GO term, however, we conducted a PubMed search of the 188 DEGs described in the PubMed literature database and noted that 63 DEGs were implicated in EndMT as either players in driving the transition or as markers. Some examples include *Transforming growth factor beta receptor 3 (Tgfbr3,* up 1.5 ×, padj = 0.01), *Collagen Type XIII α1 chain (Col13a1,* up 2.0 ×, padj = 4.2E-09), *Bone Morphogenic Protein 6 (Bmp6,* down 0.6 ×, padj = 7.0E-05), *Latent Transforming Growth Factor Binding Protein 4 (Ltbp4,* down 0.5 ×, padj = 0.008), *Connective Tissue Growth Factor (Ctgf,* down 0.7 ×, padj = 0.06), *Slit Guidance Ligand 2, (Slit2,* up 1.6 ×, padj = 2.5E-05), and α2 macroglobin (*A2m,* down 0.6 ×, 5.6E-05). Due to the results of the GO term analysis as well as the PubMed search, we speculated that *H19* might play a role in regulating EndMT.

To directly test the role of *H19* in regulating EC gene biology, we isolated primary ECs from wild-type and *H19*-deficient P2 littermates and transfected with an *H19* expression vector or with an empty control vector and then assayed gene expression after 24 hr. *H19* expression reduces expression of a mesenchymal cell marker (*SM22α*) and of genes encoding transcription factors critical for EndMT (*Snail* and *Slug*) (*Figure 5D*).

EndMT is an essential part of the normal development of many tissues/organs including heart. For example, EndMT is critical in cardiac valve development (*Kisanuki et al., 2001*; *Markwald et al., 1977*). Studies also report that EndMT contributes to cardiac diseases including cardiac fibrosis, valve calcification, and endocardial elastofibrosis (*Evrard et al., 2016*; *Goumans et al., 2008*; *Piera-Velazquez et al., 2011*; *Zeisberg et al., 2007*). During the actual EC transition, cells will transiently express endothelial markers (like CD31or IB4) simultaneously with mesenchymal markers (like aSMA or SM22a). To understand the impact of *H19* deficiency on EC transition in vivo, we fixed and sectioned hearts isolated at several developmental stages from $H19^{ΔEx1}/H19^+$ and their wild-type littermates mice and looked for co-staining of these endothelial and mesenchymal markers. At each stage, we focused on the regions of the heart where *H19* expressing cells were particularly abundant, assuming that this is where a phenotype would be most readily observed. In e14.5 embryos, we looked at endocardium, epicardium, valves, and blood vessels. In P1 embryos, we looked at endocardium, valves, and blood vessels. In adult hearts, we looked at endocardium. Comparable sections for wild-type and mutant mice were identified by a cardiac pathologist blinded to genotype before we stained for EC and mesenchymal markers. In each stage. we noted significant changes in co-staining frequency indicating that the likelihood of EC cell transition is increased in the absence of *H19* (*Figure 5E,F*). We confirmed that these results through independent analyses that compared LOI mice with their

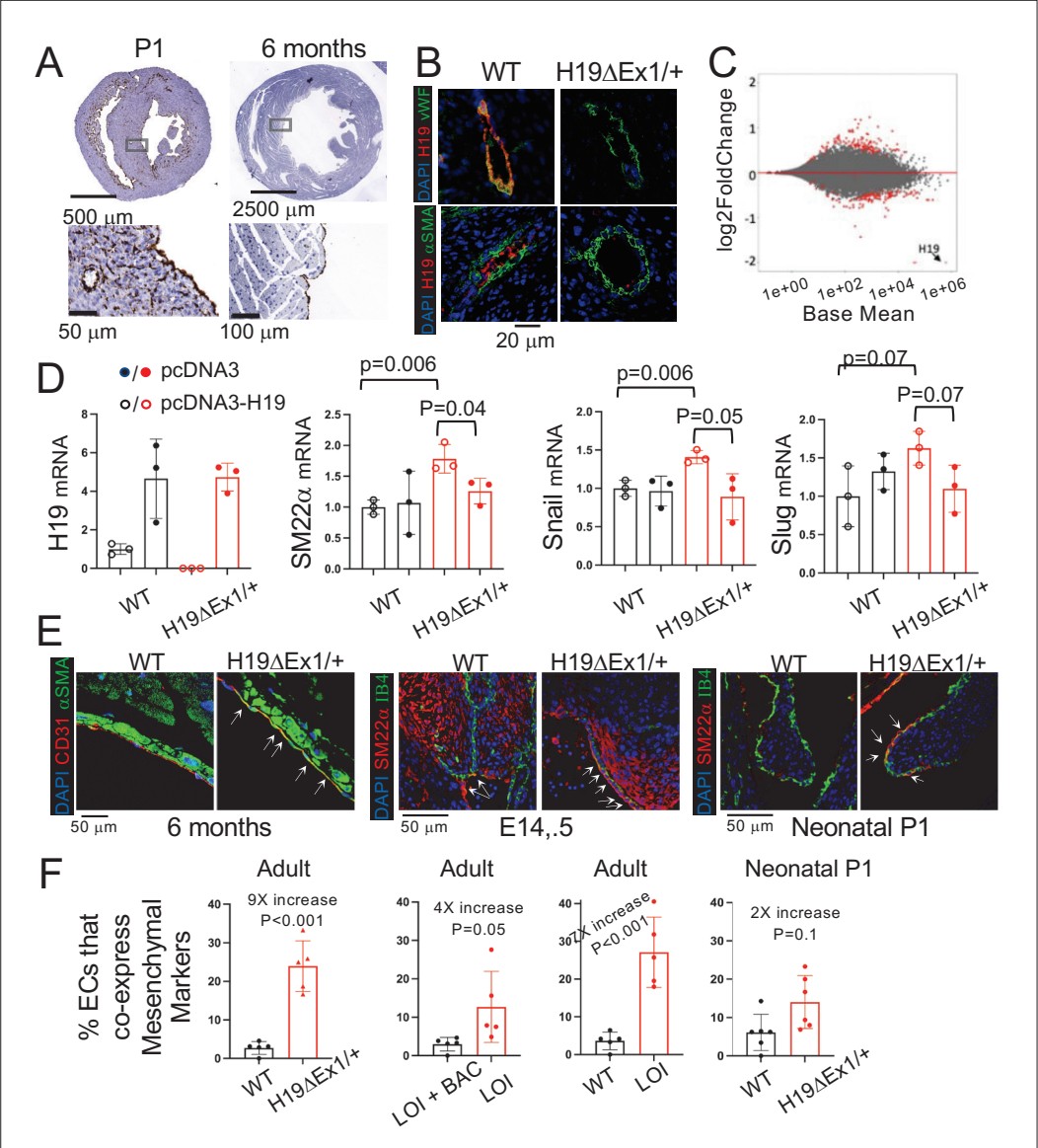

**Figure 5.** *H19* influences gene expression and cell fate in cardiac endothelial cells. (**A**) In situ staining for *H19* (brown) in hearts from wild-type P1 neonates or 6 month adults. (**B**) Combined in situ and immunohistochemistry for hearts isolated from wild-type and *H19ΔEx1/H19+* littermates at 6 months shows expression of *H19* is concentrated in endothelial cells. Sections were stained for *H19* lncRNA and then with antibodies to the endothelial marker, vWF (von Willebrand factor), or to the smooth muscle marker, α−SMA (alpha smooth muscle actin). (**C**) MA blot showing differences in expression of polyadenylated RNAs in *H19*-deficient endothelial cells. Endothelial cells were isolated from wild-type (N = 4) and *H19^ΔEx1/H19+* (N = 3) P2 neonatal hearts based on CD31 expression. RNAs were isolated and polyadenylated transcripts were quantitated. Genes marked in red are significantly differentially expressed at FDR < 0.05. Gene ontology analyses are described in **Figure 5—figure supplement 1**. (**D**) Transient transfection of *H19^ΔEx1/H19+* cardiac endothelial cells with an *H19*-expression vector rescues expression of key EndMT genes. Cardiac endothelial cells were isolated from wild-type and *H19*-deficient P2 hearts as described in (**D**) and transfected with empty expression vector (pcDNA3) or with pcDNA3 carrying mouse *H19* gDNA (pcDNA3-H19). After + hours in culture, RNA was extracted and cDNAs synthesized and analyzed for *H19*, *SM22α*, *Snail*, or *Slug*. For each gene, cDNA levels were normalized to *GAPDH* and then to the levels seen in wild-type cells transfected with pcDNA3 only. (**E, F**). Increased frequency of EndMT transitioning cells in *H19*-deficient mice. (**E**) Hearts from wild type and *H19^ΔEx1/H19+* were isolated at e14.5, P1, and at 6 months. Sections were probed for endothelial cell markers (CD31 or IB4) and for mesenchymal markers (αSMA or SM22α) to identify cells co-expressing these genes. (**F**) Frequencies of cells co-expressing endothelial and mesenchymal markers in adult and P1 hearts. The role of *H19* was determined by three independent comparisons: wild type vs.

*Figure 5 continued on next page*

*Figure 5 continued*

$H19\Delta^{Ex1}/H19^+$, LOI vs LOI+ H19 BAC, wild type vs LOI. (**D, F**). Statistical significance was evaluated using Student's t-test type 2.

The online version of this article includes the following source data and figure supplement(s) for figure 5:

**Source data 1.** H19 regulates gene expression and cell fate in cardiac endothelial cells.

**Figure supplement 1.** Visual representation of Gene Ontology analysis for endothelial cells isolated from wild-type (N = 4) and $H19^{\Delta Ex1}/H19^+$ (N = 3) P2 neonatal hearts.

wild-type littermates ($H19^{\Delta ICR}/H19+$ vs. $H19^+/H19^+$) and that compared LOI mice with LOI littermates that also carried the H19 BAC transgene $H19^{\Delta ICR}/H19^+$ vs $H19^{\Delta ICR}/H19^+$ H19 BAC Transgene (**Figure 5F**).

## Mirlet7 binding sites on the H19 lncRNA are essential for normal cardiac physiology

*H19* lncRNA is known to physically bind *Mirlet7g* (*let-7g*) and *Mirlet7i* (*let-7i*) in exon one and *Mirlet7e* (*let-7e*) in exon four in mice (**Kallen et al., 2013**). One proposed mechanism for *H19* lncRNA function is that it regulates *Mirlet7* microRNAs via these interactions to modulate their biological activities. *Mirlet7* miRNAs are known to play a role in cardiovascular diseases including cardiac hypertrophy, cardiac fibrosis, dilated cardiomyopathy, and myocardial infarction (**Bao et al., 2013**).

To test the role of *H19's Mirlet7* binding in preventing cardiomyopathy, we used CRISPR/Cas9 genome editing to delete *Mirlet7* binding sites in the *H19* gene (**Figures 1A and 6A**). Mice carrying this mutation (*H19ΔLet7/ H19ΔLet7*) express *H19* at wild-type levels (**Figure 6B**), which shows that the deletions do not disrupt lncRNA expression or stability. We used sequence-specific biotinylated oligonucleotides to purify *H19* lncRNA from extracts prepared from wild-type and from $H19^{\Delta let7}/H19^{\Delta let7}$ cells and assayed for co-purification of *Mirlet7* microRNAs. These experiments provided direct biochemical evidence that *H19* and *Mirlet7* RNAs do physically associate and also that these interactions are lost in our *H19Δlet7* deletion model (**Figure 6—figure supplement 1**).

One year old $H19^{\Delta Let7}/H19^{\Delta Let7}$ mice displayed cardiomegaly as measured by a 32 % increase in heart weight/tibia length ratios (wild type = 7.5 ± 1.7 mg/mm, N = 4; $H19^{\Delta Let7}/H19^+$ = 9.9 ± 2.2 mg/mm, N = 5; p=0.007). Transverse sections also suggested hypertrophy (**Figure 6C**), which was quantified as increased fiber diameter (**Figure 6D**). The cardiac hypertrophy in $H19^{\Delta Let7}/H19^{\Delta Let7}$ mice was accompanied by increased interstitial and perivascular fibrosis (**Figure 6E,F**). The pathologic nature of the observed cardiac myopathies in these mutant mice was confirmed by the increased levels of cardiomyopathy markers (**Figure 6G**, **Figure 6—figure supplement 2**). Thus our results support a role for *Mirlet7* miRNA binding to *H19* lncRNA in preventing cardiomyopathies.

## Discussion

BWS is an overgrowth disorder with significant patient to patient variation in disease symptoms (**Jacob et al., 2013**). An explanation for some of this variability is that independent molecular mechanisms for BWS exist (**Weksberg et al., 2010**). More than 50 % of BWS cases are associated with epigenetic lesions that disrupt expression of *CDKN1C*, an imprinted gene closely linked to *IGF2/H19* but under control of its own *ICR (IC2)*. (More rarely, BWS cases are associated with pathogenic lesions in the *CDKN1C* peptide coding sequences.) About 5 % of BWS cases are associated with disrupted imprinting at the *IGF2/H19* locus. About 20 % of cases are associated with paternal uniparental disomy of the entire region (potentially affecting both *CDKN1C* and *IGF2/H19*) (IC1 and IC2), and another 20 % of cases are of unknown origin.

There are conflicting studies regarding the correlation of artificial reproductive technology (ART) in the development of BWS. **Doornbos et al., 2007** suggested no direct effect on the increase of imprinted diseases after correcting for the increased fertility problems of the parents. However, more recent studies have examined the effect of ART on epigenetic changes (**Odom and Segars, 2010**) and suggest a 6- to 10-fold increased risk for BWS specifically because of the increased chance that *IGF2/H19* imprinting is disrupted (**Hattori et al., 2019**; **Johnson et al., 2018**; **Mussa et al., 2017**). Interestingly, BWS patients associated with ART are more likely to show cardiac problems (**Tenorio et al., 2016**), suggesting a role for *IGF2/H19* expression in normal heart development and function.

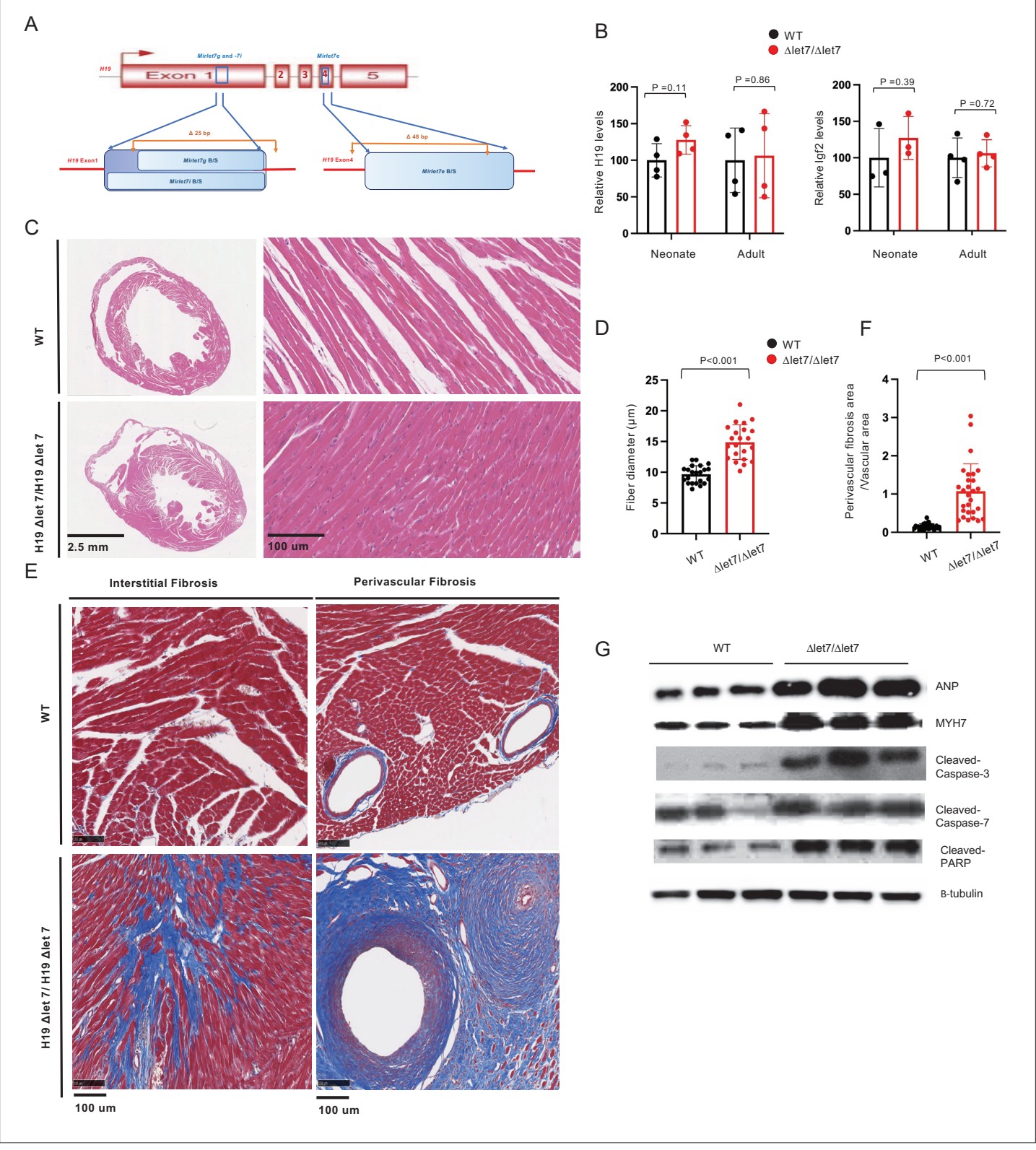

**Figure 6.** *H19*'s *Mirlet7* binding domains are essential for normal function. (**A**) The *H19ΔLet7* allele was generated by deleting 25 and 48 bp sequences within exons 1 and 4 to eliminate binding sites for *Mirlet7g*, *Mirlet7i*, and *Mirlet7e* miRNAs. (**B**) The *H19ΔLet7* allele is expressed at wild-type levels. RNAs were isolated from hearts from *H19^ΔLet7^/H19^+^* and quantitated by qRT-PCR, normalizing first to *GAPDH* and then to the levels of *H19* observed in *H19^+^/H19^+^*. Similarly, *Igf2* is expressed at equivalent levels in wild-type and mutant mice. Thus phenotypes associated with the *H19Δlet7* allele should

*Figure 6 continued*

be ascribed to changes in *H19* function and not to changes in *H19* levels or in *Igf2* expression. See *Figure 6—figure supplement 1* for experiments demonstrating that the *H19Δlet7* lncRNA cannot interact with *Mirlet7* miRNAs. (**C**) Transverse sections were collected midway along the longitudinal axis from hearts collected from 12-month-old wild-type (N = 4) and mutant (N = 3) littermates and stained with hematoxylin and eosin. (**D**) Fiber diameters were quantitated using three sections per mouse. (**E, F**). Masson's trichrome staining of sections described in panel (**C**) Red, muscle fibers; blue, collagen. Sections from three wild-type and four mutant littermates were used to calculate fibrosis. (**G**) Immunoblot analyses of whole heart extracts prepared from 12 month WT (N = 3) and mutant littermates (N = 3). Altered expression of ANP, Myh7, Cleaved Caspase-3, Cleaved Caspase-7, and Cleaved PARP. β-tubulin is a loading control. Quantified data is presented in *Figure 6—figure supplement 2*. Statistical significance was evaluated with Student's t-test type 2.

The online version of this article includes the following source data and figure supplement(s) for figure 6:

**Source data 1.** Mirlet7 binding sites on the H19lncRNA are essential to prevent cardiomyopathy.

**Figure supplement 1.** *Mirlet7e, -g, and -i miRNAs copurify with wild-type H19 lncRNA, but not with the H19Δlet7 mutant RNA.*

**Figure supplement 2.** H19-dependent changes in protein expression in adult mice – quantitated western blots.

In this study, we characterize a mouse model for *Igf2/H19* LOI. This model deletes the *Imprinting Control Region* upstream of the *H19* promoter and recapitulates the molecular phenotype of BWS patients: biallelic (i.e. 2× dosage) *IGF2* and reduced *H19* RNA. Here we show that this model not only phenocopies the transient cardiomegaly observed in neonates but also displays cardiovascular dysfunctions that are only rarely observed in patients.

To elucidate the molecular and developmental etiology of these cardiovascular phenotypes, we characterized two additional mouse models that independently altered expression of *Igf2* and of *H19*. These genetic analyses demonstrated that overexpression of *Igf2* and loss of *H19* play distinct roles in driving BWS cardiac phenotypes. In neonates, increased levels of circulating IGF2 results in hyperactivation of mTOR signaling in cardiomyocytes and thus leads to cardiomyocyte hyperplasia and cellular hypertrophy but the resultant cardiomegaly in mice is transient. As in humans, expression of *Igf2* in mice is strongly downregulated after birth and organ sizes return toward normal. Loss of *H19*, however, results in progressive cardiac pathology. Aged *H19* deficient mice show increased fibrosis, expression of markers indicative of cardiac failure, abnormal echocardiography phenotypes, low blood pressure, and aberrant vasculature. Thus, in the mouse LOI model, disease phenotypes are not restricted to fetal and neonatal stages. It will be interesting and important to assess whether this is true in other mammals.

In hearts, *H19* expression is restricted to endothelial cells. To understand the significance of *H19* expression we isolated cardiac ECs from wild-type and mutant neonates. Transcriptome analyses showed altered expression of genes associated with endothelial to mesenchymal transition, suggesting that *H19* might help regulate EC cell fate. Supporting this idea, we saw that forcing expression of *H19* in primary ECs prevents activation of mesenchymal gene expression patterns. Finally, we saw that *H19*-deficient mice show significant increases in the frequency of EC cells simultaneously expressing mesenchymal markers.

The ability of some ECs to transition of mesenchymal cells is necessary for normal development and thus can be assumed to be an essential property of ECs. The phenotypes of *H19*-deficient mice do not suggest that *H19* lncRNA is the single key molecule regulating EC cell fate. Even in LOI mice, EndMT is almost always occurring only when developmentally appropriate. Rather, our data indicate that *H19* RNA levels play a role in modulating the fate decision, so that cells lacking *H19* are modestly but measurably more likely to switch toward a transitional state where both EC and mesenchymal markers are expressed. It is interesting to note one commonality of key pathways disrupted by loss of *H19* lncRNA is that they share regulation by TGFβ signaling, suggesting that the observed 50 % reduction in expression of TGFβ receptors might be a key phenotype in *H19*-deficient ECs (*Goumans et al., 2008*).

Our findings extend earlier studies showing patterns of *H19* expression in development and in response to injury suggesting a role for *H19* in vascular physiology and pathology (*Jiang et al., 2016*; *Kim et al., 1994*). Moreover, *Voellenkle et al., 2016* recently described a role for lncRNAs including *H19* in the physiology of umbilical vein endothelial cells.

Our results also agree with in vitro studies that demonstrated an important role for *H19* in regulating EMT in cancer cells (*Li et al., 2019*; *Ma et al., 2014*; *Matouk et al., 2016*; *Matouk et al., 2014*; *Wu et al., 2019*; *Zhang et al., 2018*). In these previous analyses, *H19* function was determined by

transfecting cancer cells with *H19*-expression vectors and analyzing cell motility and gene expression. However, in contrast to our findings that activation of mesenchymal expression is associated with loss of *H19*, these in vitro analyses find EMT is induced by increasing *H19* RNA. This discrepancy emphasizes the useful role for genetic animal models in addressing developmental disorders where phenotypes are coming from cumulative changes in multiple cell types and over long periods of time. In animal models, observed phenotypes are due to the cumulative effect of the mutation in many cell types and over developmental time. In vitro studies of H19 have focused on the effect of acute changes in levels of *H19* in a single cell type.

Our analyses cannot address the fate of cells that co-express EC and mesenchymal markers. Do they all proceed toward full EndMT, do they return toward EC fates, or do they teeter in between? These questions can be addressed in future experiments using conditional *H19* deletion alleles and cell fate markers.

*H19* can be a very abundant transcript. In neonatal ECs, *H19* lncRNA represents about 1 % of all polyadenylated RNA. Yet its biochemical functions remain unclear. Various studies support the idea that *H19* functions as a microRNA precursor (*Cai and Cullen, 2007*; *Dey et al., 2014*; *Keniry et al., 2012*), as a p53 protein inhibitor (*Hadji et al., 2016*; *Park et al., 2017*; *Yang et al., 2012*; *Zhang et al., 2017*), as a regulator of DNA methylation (*Zhou et al., 2019*; *Zhou et al., 2015*), and as a modulator of *Mirlet7* microRNA functions (*Gao et al., 2014*; *Geng et al., 2018*; *Kallen et al., 2013*; *Peng et al., 2017*; *Zhang et al., 2019*; *Zhang et al., 2017*). It is possible that *H19* functions vary from cell type to cell type (*Raveh et al., 2015*). Alternatively, these functions might co-exist in a single cell but analyses to date have only looked at *H19* function from single perspectives and have missed its ability to perform in multiple pathways. To address this issue, we have begun to generate mutant *H19* alleles that disrupt specific functions. Here we show that $H19\Delta Ex1/H19^+$ have 100 × reduced levels of lncRNA but almost normal levels of *Mir675-3p and -5* p and still show cardiac pathology. Thus, the pathologies in LOI mice depend on the loss of *H19* lncRNA. To then address how the lncRNA might function, we generated mice carrying an *H19* allele missing *Mirlet7* binding sites. These mice show cardiac pathologies including extreme fibrosis. We find the fibrosis phenotype in *H19ΔLet7* mice to be especially interesting. We speculate that intensity relative to that seen in an *H19* null is most consistent with the idea that *H19* lncRNA has multiple roles in the cell and by disrupting only one role we have altered some balance, so that the animal is worse off than having no *H19* at all.

The strong phenotype in *H19ΔLet7* mice is consistent with several previous studies that emphasize the importance of *H19* lncRNA interactions with *Mirlet7* miRNAs but it is also paradoxical in that multiple studies of *Mirlet7* function in hearts indicate that *Mirlet7* functions as an anti-fibrotic factor. That is, reduced *Mirlet7* is a risk factor for fibrosis and fibrosis induces *Mirlet7*, presumably as a corrective measure (*Bao et al., 2013*; *Elliot et al., 2019*; *Sun et al., 2019*; *Wang et al., 2015*). The increased fibrosis in *H19ΔLet7* mice suggests that the simple model (that *H19* binds to and reduces *Mirlet7* bioavailability) is not correct or, more likely, that complex developmental interactions play critical roles that determine phenotypes in ways that are not yet understood. Either way, our results confirm the importance of animal models and the need for even more sophisticated conditional deletions.

*H19* and *Igf2* are generally thought of as fetal genes since their expression is so strongly repressed after birth. This fact might suggest that the adult phenotypes in *H19*-deficient mice are downstream effects of the loss of *H19* in the developing heart. However, as already mentioned, at peak expression, *H19* levels are extraordinarily high. Thus, even after 100-fold developmentally regulated decrease, *H19* remains one of the top 100 genes in terms of RNA levels. For this reason, conditional ablation models will be needed to determine exactly when *H19* expression is important.

## Materials and methods

**Key resources table**

| Reagent type (species) or resource | Designation | Source or reference | Identifiers | Additional information |
|---|---|---|---|---|
| Gene (*Mus musculus*) | H19 | GeneBank | NR_130973.1 | |
| Strain, strain background (*Mus musculus*) | C57BL/6 J (B6) | Jackson Labs | Jax: 000664 | |

*Continued on next page*

*Continued*

| Reagent type (species) or resource | Designation | Source or reference | Identifiers | Additional information |
|---|---|---|---|---|
| Strain, strain background (*Mus musculus*) | B6.FVB-Tg (Myh6-cre)2,182Mds/J | Jackson Labs | Jax: 011038 | |
| Strain, strain background (*Mus musculus*) | H19$^{\Delta ICR}$ | Previous study from this lab | PMID:10817754 | Maintained by B6 backcross |
| Strain, strain background (mouse) | H19 BAC Transgene | Previous study from this lab | PMID:10921905 | Maintained by B6 backcross |
| Strain, strain background (mouse) | H19ΔEx1 | Previous study from this lab | PMID:12270940 | Maintained by B6 backcross |
| Strain, strain background (mouse) | H19ΔLet7 | This study | See Materials and methods | Maintained by B6 backcross |
| Cell line (Rattus norvegicucs) | H9c2 cells: Rat myocardium (embryo) | ATCC CRL-1446 | RRID:CVCL_0286 | |
| Transfected construct (vector control) | pcDNA3.1+ | Invitrogen | V790-20 | |
| Biological sample (*Mus musculus*) | Primary myoblasts | This study | See Materials and methods | Isolated from neonates |
| Biological sample (*Mus musculus*) | Primary cardiac endothelial cells | This study | See Materials and methods | Freshly isolated from neonates |
| Biological sample (*Mus musculus*) | Primary cardiomycytes | This study | See Materials and methods | Freshly isolated from neonates |
| Antibody | See *Supplementary file 3* for list of 23 primary and six secondary antibodies | | | |
| Recombinant DNA reagent | pcDNA3-H19 | This study | Mouse EcoRI-SalI fragment cloned into Nhe-XhoI of pcDNA3 | |
| Sequence-based reagent | See *Supplementary file 2* for list of oligonucleotides used for PCR | | | |
| Peptide, recombinant protein | Human Recombinant IGF2 | Preprotech | Catalog # 100–12 | |
| Commercial assay or kit | Mouse IGF2 ELISA Kit | Abcam | ab100696 | |
| Commercial assay or kit | H19 in situ probe | Advanced Cell Diagnostics | ACD 423751 | |
| Commercial assay or kit | H19 in situ probe | Advanced Cell Diagnostics | ACD 322360 | |
| Chemical compound, drug | BMS 754807 | Active Biochem | A-1013 | |
| Chemical compound, drug | MEK1/2 Inhibitor PD98059 | Cell Signaling Technology | #9900 | |
| Software, algorithm | DESeq2 | | RRID:SCR_015687 | |
| Software, algorithm | Bioconductor | | RRID:SCR_006442 | |
| Software, algorithm | Feature Counts | | RRID:SCR_012919 | |
| Software, algorithm | Debian | | RRID:SCR_006638 | |
| Software, algorithm | STAR | | RRID:SCR_004463 | |

## Animal studies

All mice were bred and housed in accordance with National Institutes of Health and United States Public Health Service policies. Animal research was performed only after protocols were approved by the National Institute of Child Health and Human Development Animal Care and Use Committee.

$H19^{\Delta ICR}/H19^+$ (*Srivastava et al., 2000*) and wild-type littermates or $H19^{\Delta Ex1}/H19^+$ (*Srivastava et al., 2003*) and wild-type littermates were generated by backcrossing heterozygous females with C57BL/6 J males (Jackson Labs 000664). For tissue-specific LOI, we crossed $H19^{\Delta ICR}/H19^{ICRflox}$ females (*Srivastava et al., 2000*) with males hemizygous for the *Myh6-Cre* transgene (Jackson Labs 011038) (*Agah et al., 1997*). The H19 BAC transgene was generated as described (*Kaffer et al., 2001*; *Kaffer et al., 2000*) and used to generate $H19^{\Delta ICR}/H19^+$ *BAC+* females for backcrossing with C57BL/6 J males.

The *H19ΔLet7* allele was generated using CRISPR/Cas9 gene editing of RI mouse embryonic stem cells (ESCs). In step 1, we used gRNAs 5'-CACCGAGGGGTTGCCAGTAAAGACTG-3' and 5'-CACC GCTGCCTCCAGGGAGGTGAT-3' to delete 25 bp (AGACTGAGGCCGCTGCCTCCAGGGAGGTGAT) in exon 1. In step 2, we used gRNAs: 5'-CACCGCTTCTTGATTCAGAACGAGA-3' and 5'-CACCGACC ACTACACTACCTGCCTC-3' to delete 48 bp (CGTTCTGAATCAAGAAGATGCTGCAATCAGAACCAC TACACTACCTGC) in exon 4. Positive clones were identified by PCR screens and then confirmed by sequencing 686 bp spanning the exon 1 deletion and 1086 bp spanning the exon four deletion. Founder mice were obtained by injecting mutated ESCs into C57BL/6 J blastocyts and then back-crossed twice with C57BL/6 J females.

Genotypes were determined by PCR analyses of gDNAs extracted from ear punch biopsies (*Supplementary file 2*).

## Electrocardiography measurements

Transthoracic echocardiography was performed using a high-frequency linear array ultrasound system (Vevo 2100, VisualSonics) and the MS-400 Transducer (VisualSonics) with a center operating frequency of 30 MHz, broadband frequency of 18–38 MHz, axial resolution of 50 mm, and footprint of 20 × 5 mm. M-mode images of the left ventricle were collected from the parasternal short-axis view at the midpapillary muscles at a 90° clockwise rotation of the imaging probe from the parasternal long-axis view. Form the M-mode images, the left ventricle systolic and diastolic posterior and anterior wall thicknesses and end-systolic and -diastolic internal left ventricle chamber dimensions were measured using the leading-edge method. Left ventricular functional values of fractional shortening and ejection fraction were calculated from the wall thicknesses and chamber dimension measurements using system software. Mice were imaged in the supine position while placed on heated platform after light anesthesia using isoflurane delivered by nose cone.

## Blood pressure measurements

After sedation with isoflurane, a pressure catheter (1.0-Fr, model SPR1000, Millar Instruments, Houston, TX) was inserted into the right carotid and advanced to the ascending aorta. After 5 min acclimation, pressures were recorded using Chart 5 software (AD Instruments, Colorado Springs, CO) (*Knutsen et al., 2018*).

## Arterial pressure-diameter testing

Ascending aortas (from the root to just distal to the innominate branch point) and left carotid arteries (from the transverse aorta to 6 mm up the common carotid) were dissected and mounted on a pressure arteriograph (Danish Myotechnology, Copenhagen, Denmark) in balanced physiological saline (130 mM NaCl, 4.7 mM KCl, 1.6 mM $CaCl_2$, 1.18 mM $MgSO_4 \cdot 7H_2O$, 1.17 mM $KH_2PO_4$, 14.8 mM $NaHCO_3$, 5.5 mM dextrose, and 0.026 mM EDTA, pH 7.4) at 37 °C. Vessels were transilluminated under a microscope connected to a charge-coupled device camera and computerized measurement system (Myoview, Danish Myotechnology) to allow continuous recording of vessel diameters. Prior to data capture vessels were pressurized and stretched to in vivo length (*Wagenseil et al., 2005*). Intravascular pressure was increased from 0 to 175 mmHg in 25 mmHg steps. At each step, the outer diameter (OD) of the vessel was measured and manually recorded. Segmental distensibility was calculated from the pressure diameter curves as follows: distensibility ($SD_{25}$) over a 25 mmHg interval $= [OD_{Higher\ Pressure\ (H)} - OD_{Lower\ Pressure(L)}]/OD_{(L)}/25$ (*Knutsen et al., 2018*).

## Histological analyses

Hearts from adult mice were fixed by Langendorff perfusion or by transcardiac perfusion using 4 % paraformaldehyde (PFA) and embedded in paraffin. Fetal and neonatal hearts were isolated and then fixed by submersion in 4 % PFA before embedding. From embedded hearts, we obtained 5 mm

transverse sections for analysis. Masson's Trichrome (Sigma Aldrich, HT15, St. Louis Missouri) and Picosirius Red (Sigma Aldrich, 365548) staining were according to supplier's instructions. Fiber diameter index was quantitated using Hamamatsu-NDP software.

## Immunofluorescence and immunohistochemistry

Primary myocytes and H19C2 cells were fixed with 4 % PFA, permeabilized with 0.5 % Triton, and blocked with 10 % normal serum before incubation with antibodies. Paraffin sections were deparaffinized and rehydrated according to standard protocols. Antigen retrieval was applied using citrate buffer (Abcam, 1b93679, Cambridge, MA) for 20 min and then maintained at a sub-boiling temperature for 10 min. Sections were treated with serum-free blocking solution (DAKO, X0909, Santa Clara, CA) and all antibodies (*Supplementary file 3*) diluted in antibody diluent solution (DAKO, S0809). Secondary staining was performed for 30 min at RT. Samples were imaged with a Carl Zeiss 880 laser scanning microscope using a 40× oil immersion objective. Images were composed and edited in ZEN&LSM image software provided by Carl Zeiss or Illustrator 6.0 (Adobe).

## RNA in situ hybridization

Single color probes for H19 were purchased from Advanced Cell Diagnostics (ACD 423751, Newark, CA). RNA in-situ hybridization was performed on paraffin sections using the 2.5 HD Brown Detection Kit (ACD 322310). For dual staining with antibodies, we used H19-RD chromagen kit (ACD 322360).

## Immunoblotting

Cell extracts and tissue extracts were prepared using M-PER mammalian protein extraction buffer (Thermo Fisher 78501, Waltham, MA) or T-PER tissue protein extraction buffer (Thermo Fisher 78510), respectively. Protein concentrations were assayed using a BCA Protein Assay Kit (Pierce 23227, Waltham, MA). Proteins were fractionated by electrophoresis on 12 % or on 4%–20% SDS−PAGE gels and then transferred to nitrocellulose. Antibodies (*Supplementary file 3*) were diluted in antibody enhancer buffer (Pierce 46644).

## Cell culture

Primary cardiomyocytes were isolated form P1 pups using the Pierce Primary Cardiomyocyte Isolation Kit (Thermo Fisher 88281). H19c2 (*Branco et al., 2015*) cells were purchased from ATCC (CXRL-1446). Cells were grown at 37 °C in 5 % $CO_2$ in DMEM + 10 % FBS. Cardiomyocyte identity was confirmed by cell morphology and by immunohistochemistry for cardiomyoyte-specific markers including Myh6. Cell surface index was quantitated using Carl Zeiss-LSM software (n = 50 for each of three independent experiments).

To prepare primary endothelial cells, neonatal hearts were isolated and dissociated into single cells using Miltenyi Biotec Neonatal Heart Dissociation Kit (130-098-373, Gaithersburg, MD) but omitting the Red Cell Lysis step. Endothelial cells were purified based on CD31 expression (Miltenyi Biotec Neonatal Cardiac Endothelial Cell Isolation Kit, 130-104-183).

Primary myoblasts were generated and grown as described (*Park et al., 2017*). Cell identify is confirmed by expression of myoblast-specific markers (including MyoG), and by determining that upon serum depletion, >90% cells will differentate into elongated, multinucleate myotubes.

Cells are negative for mycoplasma contamination.

## Quantitative real-time PCR for RNA samples

Conventional RNAs were prepared from three to five independent biological samples, analyzed using a Thermo Fisher NANODROP 2000c to evaluate purity and yield, and then stored at –70 °C. cDNA samples were prepared with and without reverse transcriptase using oligo-dT primers (Roche, 04 887 352 001). cDNAs were analyzed using SYBR Green (Roche, 04 887 352 001) on the Roche Light Cycler 480 II (45 cycles with annealing at 60 °C) using primers described in *Supplementary file 2*. For each primer pair, we established standard curves to evaluate slope, y-intercepts, and PCR efficiency and to determine the dynamic range of the assay. Assay specificity was determined by melting point analyses and gel electrophoresis.

For microRNA analyses, we used mirVana™ miRNA Isolation Kit and TaqMan MicroRNA Assays Thermo Fisher, 4437975; Assay ID 001973 (U6), 001940 (Mir675-5p), 001941 (Mir675-3p).

## RNA pull-down

Primary myoblast cell cultures were generated and grown as described (*Park et al., 2017*). Cell RNA pull-down assay was performed essentially as previously described (*Lee et al., 2020*). Briefly, cells were fixed for 10 minutes at room temperature with 2.5 % formaldehyde in PBS before quenching with 0.125 M glycine for 5 min. Cells were rinsed, resuspended in NET-2 buffer (50 mM Tris–HCl, pH 7.4, 150–300 mM NaCl, 0.05% NP-40, 0.05 % deoxycholic acid, 1 mM PMSF, 2 mM benzamidine), and sonicated. Cell lysates were incubated with biotin-labeled probes synthesized by Eurofins Genomics and then pulled down with streptavidin beads (Sigma-Aldrich). Precipitated RNA and miRNA were subsequently subjected to RT-qPCR. The sequences of biotin-labeled h19 probes used in this study were as follows: CTCAGTCTTTACTGGCAACC-BIOTIN, TGTAAAATCCCTCTGGAGTC-BIOTIN, CTCCCTAGAAACTCATTCAT-BIOTIN, CTTCGAGACACCGATCACTG-BIOTIN, ATGTCATGTCTTTCTGTCAC-BIOTIN, TTGACACCATCTGTTCTTTC-BIOTIN, AAGAGGTTTACACACTCGCT-BIOTIN. The sequence of biotin-labeled control probe was CCTACGCCACCAATTTCGT-BIOTIN.

## ELISA

IGF2 secreted peptide was assayed with the Mouse IGF2 ELISA KIT (Abcam, ab100696) on 10 independent samples.

## RNA sequencing and analyses

For analyses in adult animals, RNAs were isolated from 6 month $H19^{\Delta ICR}/H19^{\Delta ICR}$ and $H19+/H19+$ littermates (two per genotype) using RNeasy Plus Mini Kit (Qiagen). Samples with RNA Integrity numbers > 9 were Ribosomal RNA depleted using RiboZero Gold Kit (Illumina). Libraries were prepared using an RNA Sample Prep V2 Kit (Illumina) and sequenced (Illumina HiSeq2500) to generate paired-end 101 bp reads that were aligned to the mouse genome version mm10 using STAR v2.5.3a (*Dobin et al., 2013*). Differential expression analyses were performed using DESeq2 (*Love et al., 2014*).

For analyses in neonates, RNAs were isolated from purified cardiac endothelial cells isolated from $H19^{\Delta Ex1}/H19^+$ (N = 3) and $H19^+/H19^+$ (N = 4) littermates. Libraries were generated from samples with RNA Integrity Numbers > 9 and were sequenced and analyzed as described above.

## Acknowledgements

We thank Victoria Biggs and Jeanne Yimdjo for animal husbandry. This work was supported by the Division of Intramural Research of the Eunice Kennedy Shriver National Institute of Child Health and Human Development.

## Additional information

### Funding

| Funder | Grant reference number | Author |
|---|---|---|
| Eunice Kennedy Shriver National Institute of Child Health and Human Development | ZIAHD001804 | Karl Pfeifer |
| National Heart, Lung, and Blood Institute | ZIA HL006247 | Beth A Kozel |

The funders had no role in study design, data collection and interpretation, or the decision to submit the work for publication.

### Author contributions

Ki-Sun Park, Conceptualization, Data curation, Formal analysis, Investigation, Methodology, Project administration, Validation, Visualization, Writing – original draft; Beenish Rahat, Conceptualization, Data curation, Formal analysis, Investigation, Methodology, Visualization, Writing - review and editing; Hyung Chul Lee, Formal analysis, Investigation, Methodology, Validation, Writing - review and editing;

Zu-Xi Yu, Formal analysis, Investigation, Validation, Writing - review and editing; Jacob Noeker, Data curation, Formal analysis, Software, Visualization, Writing – original draft; Apratim Mitra, Formal analysis, Methodology, Visualization; Connor M Kean, Claudia M Gebert, Investigation, Writing - review and editing; Russell H Knutsen, Data curation, Formal analysis, Investigation, Supervision, Visualization, Writing - review and editing; Danielle Springer, Data curation, Formal analysis, Investigation, Supervision, Writing - review and editing; Beth A Kozel, Formal analysis, Supervision, Validation, Writing - review and editing; Karl Pfeifer, Conceptualization, Formal analysis, Funding acquisition, Investigation, Project administration, Resources, Supervision, Validation, Visualization, Writing – original draft, Writing - review and editing

### Author ORCIDs
Ki-Sun Park http://orcid.org/0000-0003-1322-767X
Beenish Rahat http://orcid.org/0000-0003-0371-2356
Jacob Noeker http://orcid.org/0000-0002-4344-3114
Russell H Knutsen http://orcid.org/0000-0001-6761-6502
Danielle Springer http://orcid.org/0000-0002-6261-9744
Claudia M Gebert http://orcid.org/0000-0002-1282-1602
Beth A Kozel http://orcid.org/0000-0002-9757-7118
Karl Pfeifer http://orcid.org/0000-0002-0254-682X

### Ethics
This study was performed in strict accordance with the recommendations in the Guide for the Care and Use of Laboratory Animals of the National Institutes of Health. All animals were handled according to approved institutional animal care and use committee (IACUC) protocols (050 and 063) of the Eunice Kennedy Shriver National Institute of Child Health and Human Development. Surgery was performed under Avertin anesthesia and every effort was made to minimize suffering.

### Decision letter and Author response
Decision letter https://doi.org/10.7554/eLife.67250.sa1
Author response https://doi.org/10.7554/eLife.67250.sa2

## Additional files

### Supplementary files
• Supplementary file 1. Correlations between echocardiography phenotypes in 13 month old mice. As described in the text, each mouse presents a unique array of phenotypes. LV, left ventricular; AW, anterior wall; OT, outflow tract; sys, systole; dia, diastole.
• Supplementary file 2. Primers used for qRT-PCR and for genotyping.
• Supplementary file 3. Antibodies used in this study.
• Transparent reporting form

### Data availability
Sequencing data are deposited in the NCBI Gene Expression Omnibus (GEO) under series accession number GSE111418.

The following dataset was generated:

| Author(s) | Year | Dataset title | Dataset URL | Database and Identifier |
| --- | --- | --- | --- | --- |
| Mitra A, Park K, Yu Z, Springer D, Rahat B, Pfeifer K | 2020 | Mis-expression of Igf2 and H19 work independently on distinct cell types to cause cardiomyopathy in a Beckwtih Wiedemann mouse model | http://www.ncbi.nlm.nih.gov/geo/query/acc.cgi?acc=GSE111418 | NCBI Gene Expression Omnibus, GSE111418 |

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
