## [Decision Letter]

**Acceptance summary:**

Park et al., utilize highly tractable mouse models to interrogate a particular class of genetic lesions observed in the imprinting disorder and overgrowth syndrome, Beckwith-Wiedemann syndrome (BWS). Because more than one gene is abnormally expressed in BWS, which is caused by loss of imprinting at the H19/IGF2 locus, the authors vary the expression of both genes to investigate the source of cardiovascular phenotypes in BWS. They are able to ascribe independent cardiac phenotypes that result from IGF2 overexpression and H19 loss of expression, and provide mechanistic insight into how the altered expression of each gene contributes to cardiac dysfunction.

**Decision letter after peer review:**

Thank you for submitting your article "Cardiac pathologies in mouse loss of imprinting models are due to misexpression of H19 long noncoding RNA" for consideration by *eLife*. Your article has been reviewed by 3 peer reviewers, including Benjamin L Prosser as Reviewing Editor and Reviewer #1, and the evaluation has been overseen by Edward Morrisey as the Senior Editor.

Essential Revisions (for the authors):

1) Per the reviewer comments below, additional data is needed regarding H19 and IGF2 levels in the various models utilized, as well as validation of disruption of let7-H19 interaction or let7 levels.

2) Representative gels and images should not be relied upon for conclusive statements – please provide transparent quantification for all major observations.

*Reviewer #1 (Recommendations for the authors):*

1. Some measure of organ level hypertrophy at P1 and 2 months should be shown for LOI (and LOI+H19 BAC hearts, where appropriate).

2. In general, the cardiomegaly/hypertrophy data is not presented clearly throughout the manuscript a graph demonstrating HW/TL measurements across all genotypes at similar ages would greatly simplify and strengthen this presentation.

3. One particularly weakly supported conclusion is that LOI leads to elevated levels of cyclins or p-AKT suggested from Figure 2D, which is very unclear from the representative image (particularly once normalized to t-AKT). Other specific examples include quantification of Ki-67 or cell surface area with the loss of H19 lncRNA, or that interstitial fibrosis is increased in the LOI animals.

*Reviewer #2 (Recommendations for the authors):*

1. The authors measured serum IGF2 level as an indicator of the increased IGF2 level in LOI neonates. This is reasonable as the authors showed that the cardiac IGF2 is not cell-autonomous. As serum IGF2 is circulating, the cardiac expression pattern of InsR/Igf1r would be helpful to understand the exact target cell type of IGF2; are the IGF2 receptors expressed in a substantial way in endothelial cells, or cardiomyocytes?

2. In H19BAC mouse model, cardiac H19 expression data would be helpful to prove that the H19 expression is recovered to the wild-type level. Would the H19 expression from BAC transgene show the same pattern as the endogenous H19 expression throughout aging? Is it possible to perform H19 in situ staining on 6-month and later stages of H19BAC hearts as shown in Figure 5A?

3. In H19ΔEx1/H19+ mouse, was Igf2 expression affected? Previous mouse models with H19 deletions showed increased Igf2 expression in certain tissues. Figure 2B mentions that Igf2 is monoallelic in H19ΔEx1/H19+ neonate heart. Although it's less likely for IGF2 signaling to be altered here as cardiomegaly was not observed, it might worth confirming that the serum IGF2 level in H19ΔEx1/H19+ neonates is unchanged as the cardiac IGF2 is not cell-autonomous.

4. In line 239-240; are the heart weight/tibia length ratio values normalized to the WT littermates? According to the line 127, the value from 6-month WT mice is lower than the value from the LOI+BAC transgene mice. This doesn't match with the text in line 238-239 ("hearts from 6-month LOI mice carrying the H19 transgene are not enlarged as determined by heart weight/tibia length ratios"). Also in line 340-342 ("Adult H19ΔLet7/H19+mice displayed cardiomegaly as measured by increased heart weight/tibia length ratios"), the stage of the mice when these values were measured is missing (this wild-type value is much lower than the aforementioned 6-month wild-type value).

5. In H19ΔLet7/+ mouse, was let7 level measured in heart? If H19 functions through regulating let7 stability, it's important to confirm that let7 expression is altered in these mutants.

---

## [Author Response]

Essential Revisions (for the authors):1) Per the reviewer comments below, additional data is needed regarding H19 and IGF2 levels in the various models utilized, as well as validation of disruption of let7-H19 interaction or let7 levels.

Expression data for our 4 mouse models are fully documented:

LOI vs WT: see Figure 1C, D, E.

LOI vs LOI+H19: see Figure 2—figure supplement 2.

H19ΔEx1 vs WT: see Figure 2—figure supplement 3.

H19Δlet7 vs WT: see Figure 6 and Figure 6—figure supplement 1.

We have validated disruption of let7-H19 interactions in the H19Δlet7 model. This is described in the main text (Lines 329-33) and in Figure 6—figure supplement 6. Briefly, we used biotinylated oligonucleotides to “pull-down” H19 lncRNA and show that let-7 miRNA copurifies with wild type H19 lncRNA but not with H19Δlet7 lncRNA. Methods for this are described on lines 579-592.

2) Representative gels and images should not be relied upon for conclusive statements – please provide transparent quantification for all major observations.

The data from all western blots were quantified using Image J software and presented in Figure 2—figure supplement 1, Figure 3—figure supplement 1, and Figure 4—figure supplement 1, and Figure 6—figure supplement 2. As described in the figure legends, these supplemental figures describe quantitation of the representative blots displayed in the main figures combined with quantitation of additional blots that assayed independent biological replicates.

Reviewer #1 (Recommendations for the authors):1. Some measure of organ level hypertrophy at P1 and 2 months should be shown for LOI (and LOI+H19 BAC hearts, where appropriate).

Heart weight/tibia length data for 2 months are presented on 116-117.

2. In general, the cardiomegaly/hypertrophy data is not presented clearly throughout the manuscript a graph demonstrating HW/TL measurements across all genotypes at similar ages would greatly simplify and strengthen this presentation.

Please see above for full details. Based on the differences between different mouse models and their genetic background it would not be appropriate to compare across different data sets. To overcome this issue our study compares each mouse model with wild type littermates.

3. One particularly weakly supported conclusion is that LOI leads to elevated levels of cyclins or p-AKT suggested from Figure 2D, which is very unclear from the representative image (particularly once normalized to t-AKT). Other specific examples include quantification of Ki-67 or cell surface area with the loss of H19 lncRNA, or that interstitial fibrosis is increased in the LOI animals.

Data from experiments described in Figure 2 are quantitated and displayed in Figure 2—figure supplement. We used Image J to quantitate the data from the representative blot displayed in Figure 2D and from 2 other independent blots analyzing independent biological samples. We calculated that p-AKT and p-S6K1 increase by 1.9-fold (p = 0.01) and by 6.7-fold (p<0.001). respectively.

Reviewer #2 (Recommendations for the authors):1. The authors measured serum IGF2 level as an indicator of the increased IGF2 level in LOI neonates. This is reasonable as the authors showed that the cardiac IGF2 is not cell-autonomous. As serum IGF2 is circulating, the cardiac expression pattern of InsR/Igf1r would be helpful to understand the exact target cell type of IGF2; are the IGF2 receptors expressed in a substantial way in endothelial cells, or cardiomyocytes?

Igf1R and InsR kinase genes are each expressed in both cardiomyocytes and in cardiac endothelial cells at comparable levels. (To provide some context: in our RNA-seq data sets, counts are about 8-fold less than for GAPDH).

The importance of IGF signaling through Insr/Igf1R in cardiomyocytes and in neonatal heart development is well documented and we have added several references to the main text (Wang 2019, Li 2011, Geng 2017, Shen 2020). In fact, Shen 2020 describes the importance of IGF2 synthesized by endothelial cells in regulating embryonic cardiomyocytes which may be quite relevant in our system.

The biological functions of IGF signaling in endothelial cells are less well characterized (especially in regard to early development) but ECs clearly respond to IGF stimulation and studies of embryonic stem cells indicate that IGF signaling regulates vascular development.

Our studies show cell non-autonomous effects of LOI on cardiomyocytes but do not rule out the possibility that the 2X Igf2 dosage also impacts endothelial cells. We edited the discussion to make this point clear.

Thank you for this comment. We think the main text is significantly clearer now that we made these points explicit.

2. In H19BAC mouse model, cardiac H19 expression data would be helpful to prove that the H19 expression is recovered to the wild-type level. Would the H19 expression from BAC transgene show the same pattern as the endogenous H19 expression throughout aging? Is it possible to perform H19 in situ staining on 6-month and later stages of H19BAC hearts as shown in Figure 5A?

H19 and Igf2 expression data for H19 BAC mice are now included in Figure 2—figure supplement 2. In addition, we cite Kaffer et al., (PMID 10921905) that describes expression analyses of the BAC mice using in situ hybridization.

The patterns for expression of H19 in adult animals and their changes in regard to environmental stresses are part of an ongoing study.

3. In H19ΔEx1/H19+ mouse, was Igf2 expression affected? Previous mouse models with H19 deletions showed increased Igf2 expression in certain tissues. Figure 2B mentions that Igf2 is monoallelic in H19ΔEx1/H19+ neonate heart. Although it's less likely for IGF2 signaling to be altered here as cardiomegaly was not observed, it might worth confirming that the serum IGF2 level in H19ΔEx1/H19+ neonates is unchanged as the cardiac IGF2 is not cell-autonomous.

The reviewer brings up an interesting and important point. It is a longstanding complication for all groups analyzing H19 phenotypes that H19 deletion typically results in increased levels of Igf2. Consistent with these previous findings, H19DEx1/+ mice show levels of Igf2 mRNA that are typically 10% higher than their wild type littermates. For this reason, we were very excited to see that Igf2 is not altered in H19Dlet7/H19Dlet7 mice. This means that phenotypes in this model can be definitively assigned to changes in H19 and we can conclude that cardiomyopathy is not dependent on increased Igf2. We have modified Figure 6 to include information about Igf2 expression in H19Dlet7/H19Dlet7 mice and underline the importance of this result in the Figure 6 legend (lines 721-722).

4. In line 239-240; are the heart weight/tibia length ratio values normalized to the WT littermates? According to the line 127, the value from 6-month WT mice is lower than the value from the LOI+BAC transgene mice. This doesn't match with the text in line 238-239 ("hearts from 6-month LOI mice carrying the H19 transgene are not enlarged as determined by heart weight/tibia length ratios"). Also in line 340-342 ("Adult H19ΔLet7/H19+mice displayed cardiomegaly as measured by increased heart weight/tibia length ratios"), the stage of the mice when these values were measured is missing (this wild-type value is much lower than the aforementioned 6-month wild-type value).

We address this point in detail above. Based on the differences between genetic backgrounds, it would not be appropriate to compare between different data sets. Our study was designed so that each model is compared always to wild type littermates. Essentially, our paper describes 4 independent studies: WT vs LOI, LOI+BAC vs LOI, WT vs H19DEx1/+, and WT vs H19Dlet7/H19Dlet7. We consider it a strength of our study that we consistently account for potential strain effects and that we demonstrate H19 function in 4 completely independent comparisons.

However, we also understand the reviewer’s confusion and modified the main text to make clear that each model is being compared to its wild type littermates only.

5. In H19ΔLet7/+ mouse, was let7 level measured in heart? If H19 functions through regulating let7 stability, it's important to confirm that let7 expression is altered in these mutants.

The deletion of the let7 binding site on H19 was made with the intent to alter H19-let7 physical interactions. The model is successful as confirmed in Figure 6—figure supplement 1. In Figure 6—figure supplement 1, we also show that steady-state let7 miRNA levels are equivalent to those seen in wild type mice. Thus our conclusion is that the mutation is likely to affect let7 bioavailability. Consistent with this hypothesis, we see altered expression of several well characterized let7 target genes including Hmga2 and lpl.